# BE MORE DIVERSE THAN THE MOST DIVERSE: OPTIMAL MIXTURES OF GENERATIVE MODELS VIA MIXTURE-UCB BANDIT ALGORITHMS

**Parham Rezaei**[1]**, Farzan Farnia**[2]**, Cheuk Ting Li**[2]
[1]Sharif University of Technology, [2]The Chinese University of Hong Kong
`parham.rezaei@sharif.edu, farnia@cse.cuhk.edu.hk, ctli@ie.cuhk.edu.hk`

## ABSTRACT

The availability of multiple training algorithms and architectures for generative models requires a selection mechanism to form a single model over a group of well-trained generation models. The selection task is commonly addressed by identifying the model that maximizes an evaluation score based on the diversity and quality of the generated data. However, such a best-model identification approach overlooks the possibility that a mixture of available models can outperform each individual model. In this work, we numerically show that a mixture of generative models on benchmark image datasets can indeed achieve a better evaluation score (based on FID and KID scores), compared to the individual models. This observation motivates the development of efficient algorithms for selecting the optimal mixture of the models. To address this, we formulate a quadratic optimization problem to find an optimal mixture model achieving the maximum of kernel-based evaluation scores including kernel inception distance (KID) and Rényi kernel entropy (RKE). To identify the optimal mixture of the models using the fewest possible sample queries, we view the selection task as a multi-armed bandit (MAB) problem and propose the *Mixture Upper Confidence Bound (Mixture-UCB)* algorithm that provably converges to the optimal mixture of the involved models. More broadly, the proposed Mixture-UCB can be extended to optimize every convex quadratic function of the mixture weights in a general MAB setting. We prove a regret bound for the Mixture-UCB algorithm and perform several numerical experiments to show the success of Mixture-UCB in finding the optimal mixture of text and image generative models. The project code is available at `https://github.com/Rezaei-Parham/Mixture-UCB`.

## 1 INTRODUCTION

The rapid advancements in generative modeling have created a need for mechanisms to combine multiple well-trained generative models, each developed using different algorithms and architectures, into a single unified model. Consider $m$ unconditional generative models $\mathcal{G}_1, \ldots, \mathcal{G}_m$, where each $\mathcal{G}_i$ represents a probability model $P_{\mathcal{G}_i}$ according to which new samples are generated. A common approach for creating a unified model is to compute evaluation scores (e.g., the standard FID (Heusel et al., 2017) and KID (Bińkowski et al., 2018) scores) that quantify the diversity and fidelity of the generated data, followed by selecting the model $P_{\mathcal{G}_{i^*}}$ with the best evaluated score. This best-score model selection strategy has been widely adopted for choosing generative models across various domains, including image, text, and video data generation.

However, the model selection approach by identifying the score-maximizing model overlooks the possibility that a mixture of the generative models $\alpha_1 P_{\mathcal{G}_1} + \cdots + \alpha_m P_{\mathcal{G}_m}$, where each sample is generated from a randomly-selected model with $\alpha_i$ being the probability of selecting model $\mathcal{G}_i$, can outperform every individual model. This motivates the following question: Can there be real-world settings where a non-degenerate mixture of some well-trained generative models obtain a better evaluation score compared to each individual model? Note that the standard FID and KID scores are convex functions of the generative model's distribution, and thus they can be optimized by a non-degenerate mixture of the models. In this work, we numerically show that it is possible for a

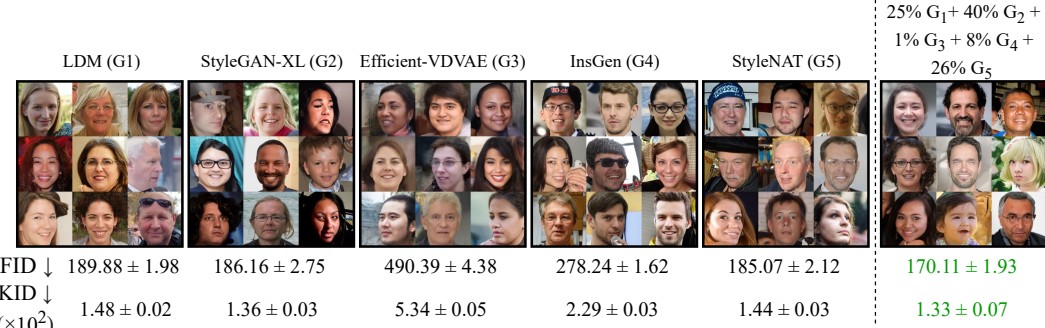

| | LDM (G1) | StyleGAN-XL (G2) | Efficient-VDVAE (G3) | InsGen (G4) | StyleNAT (G5) | 25% $G_1$+ 40% $G_2$ + 1% $G_3$ + 8% $G_4$ + 26% $G_5$ |
|---|---|---|---|---|---|---|
| FID ↓ | 189.88 ± 1.98 | 186.16 ± 2.75 | 490.39 ± 4.38 | 278.24 ± 1.62 | 185.07 ± 2.12 | 170.11 ± 1.93 |
| KID ↓ (×10²) | 1.48 ± 0.02 | 1.36 ± 0.03 | 5.34 ± 0.05 | 2.29 ± 0.03 | 1.44 ± 0.03 | 1.33 ± 0.07 |

Figure 1: A mixture (right-most case) of FFHQ pre-trained generative models with weights (0.25,0.4,0.01,0.08,0.26) achieves better FID and KID scores compared to each of the five involved models. The mixture weights are computed using our proposed Mixture-UCB-OGD algorithm.

mixture of real-world generative models to improve evaluation scores over the individual models. An example is shown in Figure 1, where we find a non-degenerate mixture of five generative models pre-trained on the FFHQ dataset[1]. As shown, assigning mixture weights $(0.25, 0.4, 0.01, 0.08, 0.26)$ to the models results in a significantly better FID score $170.11$ than the best individual FID $185.07$.

To understand the improvement achieved by the mixture model, we note that the FID and KID scores evaluate both the quality and diversity of the generated data. While the averaged quality of generated samples represents an expected value that is optimized by an individual model, the diversity of samples from a mixture of the models can significantly improve over the diversity of the individual models' data. Figure 2 displays an illustrative example for this point, where we observe that the diversity of "red bird, cartoon style" samples generated by each of the three text-to-image models, is qualitatively and quantitatively[2] lower than the diversity of their mixture. As a result, the improvement in the diversity of a mixture of generative models can result in an improved FID and KID evaluation scores, as we numerically observe in Figure 1.

## 1.1 COMPUTING OPTIMAL MIXTURES OF GENERATIVE MODELS: THE MIXTURE-UCB MULTI-ARMED BANDIT ALGORITHM

Since the evaluation score of a mixture of generative models can improve over the scores of the individual models, a natural question is how to efficiently compute the weights of an optimal mixture of the models using the fewest possible samples from the models. Here, our goal is to minimize the number of sample generation queries from sub-optimal models, which will save the time and monetary costs of identifying the best model. To achieve this, we propose viewing the task as a multi-armed bandit (MAB) problem, in which every generative model represents an arm and our goal is to find the best mixture of the models with the optimal evaluation score. The MAB approach for selecting among generative models has been recently explored by Hu et al. (2024) applying the *Upper Confidence Bound (UCB)* algorithm to the FID score. This MAB-based model selection extends the online model selection methods for supervised models, including the *successive halving* strategy (Karnin et al., 2013; Jamieson & Talwalkar, 2016; Chen & Ghosh, 2024)[3].

However, in the existing MAB algorithms, the goal is to eventually converge to a *single* arm with the optimal score. Successive halving (Karnin et al., 2013; Jamieson & Talwalkar, 2016; Chen & Ghosh, 2024) and the UCB algorithm developed by Hu et al. (2024) will ultimately select only one generative model after a sufficient number of iterations. However, as we discussed earlier, the evaluation scores of the generated data could be higher when the sample generation follows a mixture of models rather than a single model. This observation leads to the following task: Developing an MAB algorithm that finds the optimal mixture of the arms rather than the single best arm.

---

[1] The pre-trained generative models are downloaded from dgm-eval GitHub repository (Stein et al., 2023).

[2] We have evaluated the diversity scores RKE (Jalali et al., 2023) and Vendi (Dan Friedman & Dieng, 2023).

[3] Jamieson & Talwalkar (2016) focused on applying successive halving on hyperparameter optimization for supervised learning, whereas Chen & Ghosh (2024) focused on generative models using maximum mean discrepancy (Gretton et al., 2012) as the score.

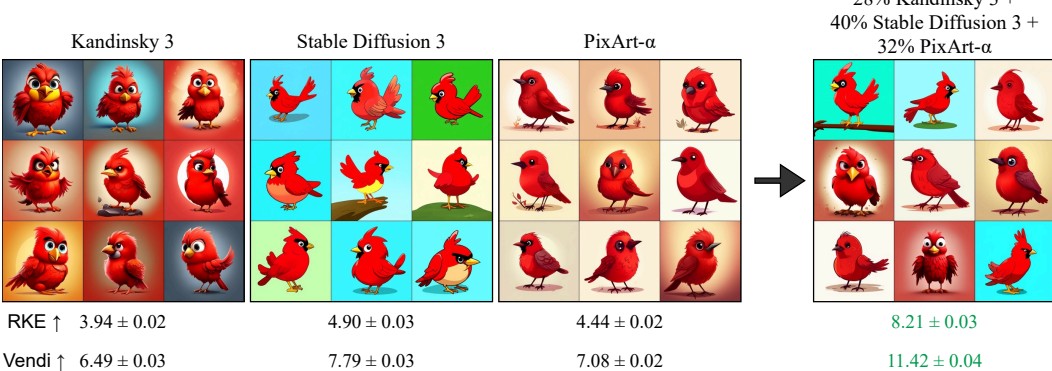

| | Kandinsky 3 | Stable Diffusion 3 | PixArt-α | 28% Kandinsky 3 + 40% Stable Diffusion 3 + 32% PixArt-α |
|---|---|---|---|---|
| RKE ↑ | $3.94 \pm 0.02$ | $4.90 \pm 0.03$ | $4.44 \pm 0.02$ | $8.21 \pm 0.03$ |
| Vendi ↑ | $6.49 \pm 0.03$ | $7.79 \pm 0.03$ | $7.08 \pm 0.02$ | $11.42 \pm 0.04$ |

Figure 2: Visual comparison of the diversity across individual arms and the optimal mixture for images generated using models Kandinsky 3, Stable Diffusion 3, and PixArt-$\alpha$ with the prompt "Red bird, cartoon style". The mixture weights are computed via the Mixture-UCB-OGD method.

In this work, we develop a general MAB method, which we call the *Mixture-UCB* algorithm, that can provably find the best mixture of the arms when the evaluation score is a quadratic function of the arms' distribution, i.e. when it represents the average of scores assigned to the pairs of drawn samples. Formulating the optimization problem for a quadratic score function results in a quadratic online convex optimization problem that can be efficiently solved using the online gradient descent algorithm. More importantly, we establish a concentration bound for the quadratic function of the mixture weights, which enables us to extend the UCB algorithm to the Mixture-UCB method for the online selection of the mixture weights.

For selecting mixtures of generative models using Mixture-UCB, we focus on evaluation scores that reduce to a quadratic function of the generative model's distribution, including Kernel Inception Distance (KID) (Bińkowski et al., 2018), Rényi Kernel Entropy (RKE) (Jalali et al., 2023) scores, as well as the quality-measuring Precision (Sajjadi et al., 2018; Kynkäänniemi et al., 2019) and Density (Naeem et al., 2020) scores, which are linear functions of the generative distribution. Among these scores, RKE provides a reference-free entropy function for assessing the diversity of generated data, making it suitable for quantifying the variety of generated samples. Our mixture-based MAB framework can therefore be applied to find the mixture model with the maximum RKE-based diversity score. Additionally, we consider a linear combination of RKE with the Precision, Density, or KID quality scores to find a mixture of models that offers the best trade-off between quality and diversity.

We perform several numerical experiments to test the application of our proposed Mixture-UCB approach in comparison to the Vanilla-UCB and One-Arm Oracle approaches that tend to generate samples from only one of the available generative models. Our numerical results indicate that the Mixture-UCB algorithms can generate samples with higher RKE diversity scores, and tends to generate samples from a mixture of several generative models when applied to image-based generative models. Also, we test the performance of Mixture-UCB on the KID, Precision, and Density scores, which similarly result in a higher score value for the mixture model found by the Mixture-UCB algorithm. We implement the Mixture-UCB by solving the convex optimization sub-problem at every iteration and also by applying the online gradient descent algorithm at every iteration. In our experiments, both implementations result in satisfactory results and can improve upon learning strategies tending to select only one generative model. Here is a summary of this work's contributions:

- Studying the selection task for mixtures of multiple generative models to improve the evaluation scores of generated samples (Section 4).

- Proposing an online learning multi-armed bandit framework to address the mixture selection task for quadratic score functions (Section 5).

- Developing the Mixture-UCB-CAB and Mixture-UCB-OGD algorithms to solve the formulated online learning problem and proving a regret bound for Mixture-UCB-CAB (Sections 5.1, 5.2).

- Presenting numerical results on the improvements in the diversity of generated data by the online selection of a mixture of the generation models (Section 6, Appendix 8.4).

## 2 RELATED WORK

**Assessment of Generative Models**. The evaluation of generative models has been extensively studied, with a focus on both diversity and quality of generated images. Reference-free metrics such as Rényi Kernel Entropy (RKE) (Jalali et al., 2023) and VENDI (Dan Friedman & Dieng, 2023; Ospanov et al., 2024) measure diversity without relying on ground-truth, while reference-based metrics such as Recall (Sajjadi et al., 2018; Kynkäänniemi et al., 2019) and Coverage (Naeem et al., 2020) assess diversity relative to real data. For image quality evaluation, Density and Precision metrics (Naeem et al., 2020; Kynkäänniemi et al., 2019) provide measures based on alignment with a reference distribution. The Wasserstein distance (Arjovsky et al., 2017) and Fréchet Inception Distance (FID) (Heusel et al., 2017) approximate the distance between real and generated datasets, while Kernel Inception Distance (KID) (Bińkowski et al., 2018) uses squared maximum mean discrepancy for a kernel-based comparison of distributions. Wang et al. (2023) apply the KID score for the distributed evaluation of generative models. The novelty evaluation of generative models has also been studied in References (Han et al., 2022; Jiralerspong et al., 2023; Zhang et al., 2024b;a).

**Multi-Armed Bandit Algorithms.** The Multi-Armed Bandit (MAB) problem is a foundational topic in reinforcement learning, where an agent aims to maximize rewards from multiple options (arms) with initially unknown reward distributions (Lai & Robbins, 1985; Thompson, 1933). The Upper Confidence Bound (UCB) algorithm (Agrawal, 1995a; Auer, 2002; Bubeck et al., 2012) is a widely adopted method for addressing the MAB problem, where uncertainty about an arm's reward is replaced by an optimistic estimate. In generative models, optimism-based bandits have been applied to efficiently identify models with optimal Fréchet Inception Distance (FID) or Inception Score while minimizing data queries (Hu et al., 2024). A special case of MAB, the continuum-armed bandit (CAB) problem (Agrawal, 1995b), optimizes a function over continuous inputs, and has been applied to machine learning tasks such as hyperparameter optimization (Feurer & Hutter, 2019; Li et al., 2018). Recent research explores CABs under more general smoothness conditions like Besov spaces (Singh, 2021), while other works have focused on regret bounds and Lipschitz conditions (Kleinberg, 2004; Kleinberg et al., 2019; Bubeck et al., 2008).

Another related reference is informational multi-armed bandits (Weinberger & Yemini, 2023), which extends UCB to maximizing the Shannon entropy of a discrete distribution. In comparison, the algorithms in this paper can minimize the expectation of any quadratic positive-semidefinite function, which also covers the order-2 Rényi entropy for discrete distributions. Since the generative models' outputs are generally continuous, (Weinberger & Yemini, 2023) is not applicable to our setting.

## 3 PRELIMINARIES

We review several kernel-based performance metrics of generative models.

### 3.1 RÉNYI KERNEL ENTROPY

The *Rényi Kernel Entropy* (Jalali et al., 2023) of the distribution $P$, which measures the diversity of the modes in $P$, is given by $\log(1/\mathbb{E}_{X,X' \overset{\text{iid}}{\sim} P}[k^2(X, X')])$, where $k$ is a positive definite kernel.[4]

Taking the exponential of the Rényi Kernel Entropy, we have the *RKE mode count* $1/\mathbb{E}[k^2(X, X')])$ (Jalali et al., 2023), which is an estimate of the number of modes. Maximizing the RKE mode count is equivalent to minimizing the following loss

$$\mathbb{E}_{X,X' \overset{\text{iid}}{\sim} P}[k^2(X, X')]. \tag{1}$$

### 3.2 MAXIMUM MEAN DISCREPANCY AND KERNEL INCEPTION DISTANCE

The (squared) *maximum mean discrepancy* (MMD) (Gretton et al., 2012) between distributions $P, Q$, which measures the distance between $P$ and $Q$, can be written as

$$\mathbb{E}[k(X, X')] + \mathbb{E}[k(Y, Y')] - 2\mathbb{E}[k(X, Y)], \tag{2}$$

---

[4]The order-2 Rényi entropy for discrete distributions is a special case by taking $k(x, x') = \mathbf{1}_{x=x'}$.

where $X, X' \overset{\text{iid}}{\sim} P$ and $Y, Y' \overset{\text{iid}}{\sim} Q$, and $k$ is a positive definite kernel. Suppose $P$ is the distribution of samples from a generative model, and $Q$ is a reference distribution. Minimizing the MMD can ensure that $P$ is close to $Q$. The *Kernel Inception Distance* (KID) (Bińkowski et al., 2018), a popular quality metric for image generative models, is obtained by first passing $P$ and $Q$ through the Inception network (Szegedy et al., 2016), and then computing their MMD, i.e., we have

$$\mathbb{E}[k(\psi(X), \psi(X'))] + \mathbb{E}[k(\psi(Y), \psi(Y'))] - 2\mathbb{E}[k(\psi(X), \psi(Y))], \tag{3}$$

where $\psi$ is the mapping from $x$ to its Inception representation.

## 4 OPTIMAL MIXTURES OF GENERATIVE MODELS

RKE (1), MMD (2) and KID (3) can all be written as a loss function in the following form

$$L(P) := \mathbb{E}_{X, X' \overset{\text{iid}}{\sim} P}[\kappa(X, X')] + \mathbb{E}_{X \sim P}[f(X)], \tag{4}$$

where $\kappa : \mathcal{X}^2 \to \mathbb{R}$ is a positive semidefinite kernel, and $f : \mathcal{X} \to \mathbb{R}$ is a function. For (1), we take $\kappa(x, x') = k^2(x, x')$ (the square of a kernel is still a kernel) and $f(x) = 0$. For (2), we take $\kappa(x, x') = k(x, x')$ and $f(x) = -2\mathbb{E}_{Y \sim Q}[k(x, Y)]$ (the constant term $\mathbb{E}[k(Y, Y')]$ does not matter). For KID (3), we take $\kappa(x, x') = k(\psi(x), \psi(x'))$ and $f(x) = -2\mathbb{E}_{Y \sim Q}[k(\psi(x), \psi(Y))]$. Note that any convex combinations of (1), (2) and (3) is still in the form (4).

Suppose we are given $m$ generative models, where model $i$ generates samples from the distribution $P_i$. If our goal is merely to find the model that minimize the loss (4), we should select $\text{argmin}_i L(P_i)$. Nevertheless, for diversity metrics such as RKE, it is possible that a mixture of the models will give a better diversity. Assume that the mixture weight of model $i$ is $\alpha_i \in [0, 1]$, where $\boldsymbol{\alpha} = (\alpha_1, \ldots, \alpha_m)$ is a probability vector. The loss of the mixture distribution $\sum_{i=1}^{m} \alpha_i P_i$ can then be expressed as

$$L(\boldsymbol{\alpha}) := L\Big(\sum_{i=1}^{m} \alpha_i P_i\Big) = \boldsymbol{\alpha}^\top \mathbf{K} \boldsymbol{\alpha} + \mathbf{f}^\top \boldsymbol{\alpha},$$

$$\mathbf{K} := (\mathbb{E}_{X \sim P_i, X' \sim P_j}[\kappa(X, X')])_{i,j \in [m]} \in \mathbb{R}^{m \times m}, \quad \mathbf{f} := (\mathbb{E}_{X \sim P_i}[f(X)])_{i=1}^{m} \in \mathbb{R}^m.$$

Given $\mathbf{K}, \mathbf{f}$, the probability vector $\boldsymbol{\alpha}$ minimizing $L(\boldsymbol{\alpha})$ can be found via a convex quadratic program.

In practice, we do not know the precise $\mathbf{K}, \mathbf{f}$, and have to estimate them using samples. Suppose we have the samples $x_{i,1}, \ldots, x_{i,n_i}$ from the distribution $P_i$ for $i = 1, \ldots, m$, where $n_i$ is the number of observed samples from model $i$. Write $\mathbf{x} := (x_{i,a})_{i \in [m], a \in [n_i]}$. We approximate the true mixture distribution $\sum_{i=1}^{m} \alpha_i P_i$ by the empirical mixture distribution $\sum_{i=1}^{m} \frac{\alpha_i}{n_i} \sum_{a=1}^{n_i} \delta_{x_{i,a}}$, where we assign a weight $\alpha_i/n_i$ to samples $x_{i,a}$ from model $i$, and $\delta_{x_{i,a}}$ denotes the degenerate distribution at $x_{i,a}$. We then approximate $L(\boldsymbol{\alpha})$ by the sample loss

$$\hat{L}(\boldsymbol{\alpha}; \mathbf{x}) := L\Big(\sum_{i=1}^{m} \frac{\alpha_i}{n_i} \sum_{a=1}^{n_i} \delta_{x_{i,a}}\Big) = \boldsymbol{\alpha}^\top \hat{\mathbf{K}}(\mathbf{x}) \boldsymbol{\alpha} + \hat{\mathbf{f}}(\mathbf{x})^\top \boldsymbol{\alpha}, \tag{5}$$

$$\hat{\mathbf{K}}(\mathbf{x}) := \Big(\frac{1}{n_i n_j} \sum_{a=1}^{n_i} \sum_{b=1}^{n_j} \kappa(x_{i,a}, x_{j,b})\Big)_{i,j} \in \mathbb{R}^{m \times m}, \quad \hat{\mathbf{f}}(\mathbf{x}) := \Big(\frac{1}{n_i} \sum_{a=1}^{n_i} f(x_{i,a})\Big)_{i=1}^{m} \in \mathbb{R}^m.$$

The minimization of $\hat{L}(\boldsymbol{\alpha}; \mathbf{x})$ over probability vectors $\boldsymbol{\alpha}$ is still a convex quadratic program.

## 5 ONLINE SELECTION OF OPTIMAL MIXTURES – MIXTURE MULTI-ARMED BANDIT

Suppose we are given $m$ generative models, but we do not have any prior information about them. Our goal is to use these models to generate a collection of samples $(x^{(t)})_{i \in [T]}$ in $T$ rounds that minimizes the loss (4) $L(\hat{P}^{(T)})$ at the empirical distribution $\hat{P}^{(T)} = T^{-1} \sum_{t=1}^{T} \delta_{x^{(t)}}$. We have

$$L(\hat{P}^{(T)}) = \frac{1}{T^2} \sum_{s,t \in [T]} \kappa(x^{(s)}, x^{(t)}) + \frac{1}{T} \sum_{t=1}^{T} f(x^{(t)}).$$

---

**Algorithm 1** Mixture-UCB-CAB

---

1: **Input:** $m$ generative arms, number of rounds $T$
2: **Output:** Gathered samples $\mathbf{x}^{(T)}$
3: **for** $t \in \{0, \ldots, m-1\}$ **do**
4:     Pull arm $t+1$ at time $t+1$ to obtain sample $x_{t+1,1} \sim P_{t+1}$. Set $n_{t+1}^{(m)} = 1$.
5: **end for**
6: **for** $t \in \{m, \ldots, T-1\}$ **do**
7:     Compute an estimate of the optimal mixture distribution via the convex quadratic program:

$$\boldsymbol{\alpha}^{(t)} := \operatorname{argmin}_{\boldsymbol{\alpha}} \big( \hat{L}(\boldsymbol{\alpha}; \mathbf{x}^{(t)}) - (\boldsymbol{\epsilon}^{(t)})^\top \boldsymbol{\alpha} \big), \tag{6}$$

    where the minimization is over probability vectors $\boldsymbol{\alpha}$, and $\boldsymbol{\epsilon}^{(t)} \in \mathbb{R}^m$ is defined as

$$\epsilon_i^{(t)} := \Delta_L \sqrt{(\beta \log t)/(2n_i^{(t)})} + \Delta_\kappa / n_i^{(t)}. \tag{7}$$

8:     Generate the arm index $b^{(t+1)} \in [m]$ at random with $\mathbb{P}(b^{(t+1)} = i) = \alpha_i^{(t)}$.
9:     Pull arm $b = b^{(t+1)}$ at time $t+1$ to obtain a new sample $x_{b,n_b^{(t)}+1} \sim P_b$. Set $n_b^{(t+1)} = n_b^{(t)} + 1$
    and $n_j^{(t+1)} = n_j^{(t)}$ for $j \neq b$.[5]
10: **end for**
11: **return** samples $\mathbf{x}^{(T)}$

---

If we are told by an oracle the optimal mixture $\boldsymbol{\alpha}^*$ that minimizes the loss $L(\boldsymbol{\alpha})$, then we should generate samples according to this mixture distribution, giving $\approx \alpha_i^* T$ samples from model $i$. We call this the *mixture oracle* scenario. Nevertheless, in reality, we do not know $\mathbf{K}, \mathbf{f}$, and cannot compute $\boldsymbol{\alpha}^*$ exactly. Instead, we have to approximate $\boldsymbol{\alpha}^*$ by minimizing the sample loss $\hat{L}(\boldsymbol{\alpha}; \mathbf{x})$ (5). However, we do not have the samples $\mathbf{x}$ at the beginning in order to compute $\hat{L}(\boldsymbol{\alpha}; \mathbf{x})$, so we have to generate some samples first. Yet, to generate these initial samples, we need an estimate of $\boldsymbol{\alpha}^*$, or else those samples may have a suboptimal empirical distribution and affect our final loss $L(\hat{P}^{(T)})$, or we will have to discard those initial samples which results in wastage.

This "chicken and egg" problem is naturally solved by an online learning approach via multi-armed bandit. At time $t = 1, \ldots, T$, we choose and pull an arm $b^{(t)} \in [m]$ (i.e., generate a sample from model $b^{(t)}$), and obtain a sample $x^{(t)}$ from the distribution $P_{b^{(t)}}$. The choice $b^{(t)}$ can depend on all previous samples $x^{(1)}, \ldots, x^{(t-1)}$. Unlike conventional multi-armed bandit where the goal is to maximize the total reward over $T$ rounds, here we minimize the loss $L(\hat{P}^{(T)})$ which involve cross terms $\kappa(x^{(s)}, x^{(t)})$ between samples at different rounds. Note that if $\kappa(x, x') = 0$, then this reduces to the conventional multi-armed bandit setting by taking $f(x)$ to be the negative reward of the sample $x$. In the following subsections, we will propose two new algorithms that are generalizations of the upper confidence bound (UCB) algorithm for multi-armed bandit (Agrawal, 1995a; Auer, 2002).

## 5.1 MIXTURE UPPER CONFIDENCE BOUND – CONTINUUM-ARMED BANDIT

Let $n_i^{(t)}$ be the number of times arm $i$ has been pulled up to time $t$. Let $\mathbf{x}^{(t)} := (x_{i,a})_{i \in [m], a \in [n_i^{(t)}]}$ be the observed samples up to time $t$. We focus on bounded loss functions, and assume that $\kappa : \mathcal{X}^2 \to [\kappa_0, \kappa_1]$ and $f : \mathcal{X} \to [f_0, f_1]$ are bounded. Let $\Delta_\kappa := \kappa_1 - \kappa_0$, $\Delta_f := f_1 - f_0$. Define the *sensitivity* of $L$ as $\Delta_L := 2\Delta_\kappa + \Delta_f$.

We now present the *mixture upper confidence bound – continuum-armed bandit (Mixture-UCB-CAB) algorithm*. It has a parameter $\beta > 1$. It treats the online selection problem as multi-armed bandit with infinitely many arms similar to the continuum-armed bandit settings in (Agrawal, 1995b; Lu et al., 2019). Each arm is a probability vector $\boldsymbol{\alpha}$. By pulling the arm $\boldsymbol{\alpha}$, we generate a sample from a randomly chosen model, where model $i$ is chosen with probability $\alpha_i$. Refer to Algorithm 1.

Similar to UCB, Mixture-UCB-CAB finds a lower confidence bound $\hat{L}(\boldsymbol{\alpha}; \mathbf{x}^{(t)}) - (\boldsymbol{\epsilon}^{(t)})^\top \boldsymbol{\alpha}$ of the true loss $L(\boldsymbol{\alpha})$ at each round. To justify the expressions (6), (7), we prove that $\hat{L}(\boldsymbol{\alpha}; \mathbf{x}^{(t)}) - (\boldsymbol{\epsilon}^{(t)})^\top \boldsymbol{\alpha}$ in (6) lower-bounds $L(\boldsymbol{\alpha})$ with probability at least $1 - t^{-\beta}$. The proof is given in Appendix 8.1.

**Theorem 1** *Fix a probability vector $\boldsymbol{\alpha}$.[6] Suppose we have samples $x_{i,1}, \ldots, x_{i,n_i}$ from the distribution $P_i$ for $i = 1, \ldots, m$, where $n_i$ is the number of observed samples from model $i$. For $\delta > 0$,*

$$\mathbb{P}\big(\hat{L}(\boldsymbol{\alpha}; \mathbf{x}) - L(\boldsymbol{\alpha}) \geq \boldsymbol{\epsilon}(\delta)^\top \boldsymbol{\alpha}\big) \leq \delta, \quad \mathbb{P}\big(L(\boldsymbol{\alpha}) - \hat{L}(\boldsymbol{\alpha}; \mathbf{x}) \geq \boldsymbol{\epsilon}(\delta)^\top \boldsymbol{\alpha}\big) \leq \delta,$$

*where $\boldsymbol{\epsilon}(\delta) := \big(\Delta_L \sqrt{\frac{\log(1/\delta)}{2n_i}} + \frac{\Delta_\kappa}{n_i}\big)_{i \in [m]}$.*

We now prove that Mixture-UCB-CAB gives an expected loss $\mathbb{E}[L(\hat{P}^{(T)})]$ that converges to the optimal loss $\min_{\boldsymbol{\alpha}} L(\boldsymbol{\alpha})$ by bounding their gap. This means that Mixture-UCB-CAB is a zero-regret strategy by treating $\mathbb{E}[L(\hat{P}^{(T)})] - \min_{\boldsymbol{\alpha}} L(\boldsymbol{\alpha})$ as the average regret per round.[7] The proof is given in Appendix 8.2.

**Theorem 2** *Suppose $m \geq 2$, $\beta \geq 4$. Consider bounded quadratic loss function (4) with $\kappa$ being positive semidefinite. Let $\hat{P}^{(T)}$ be the empirical distribution of the first $T \geq 2$ samples $\mathbf{x}^{(T)}$ given by Mixture-UCB-CAB. Then the gap between the expected loss and the optimal loss is bounded by*

$$\mathbb{E}\big[L(\hat{P}^{(T)})\big] - \min_{\boldsymbol{\alpha}} L(\boldsymbol{\alpha}) \leq 4\Delta_L \sqrt{\frac{\beta m \log T}{T}}.$$

When $\kappa(x, x') = 0$, Mixture-UCB-CAB reduces to the conventional UCB, and Theorem 2 coincides with the $O(\sqrt{(m \log T)/T})$ distribution-free bound on the regret per round of conventional UCB (Bubeck et al., 2012). Since there is a $\Omega(\sqrt{m/T})$ minimax lower bound on the regret per round even for conventional multi-armed bandit without the quadratic kernel term (Bubeck et al., 2012, Theorem 3.4), Theorem 2 is tight up to a logarithmic factor.

The main difference between Mixture-UCB-CAB and conventional UCB is that we choose a mixture of arms in (6) given by the probability vector $\boldsymbol{\alpha}$, instead of a single arm. A more straightforward application of UCB would be to simply find the single arm that minimizes the lower bound in (6), i.e., we restrict $\boldsymbol{\alpha} = \mathbf{e}_i$ for some $i \in [m]$, where $\mathbf{e}_i$ is the $i$-th basis vector, and minimize (6) over $i$ instead. We call this *Vanilla-UCB*. Vanilla-UCB fails to take into account the possibility that a mixture may give a smaller loss than every single arm. In the long run, Vanilla-UCB converges to pulling the best single arm instead of the optimal mixture. Vanilla-UCB will be used as a baseline to be compared with Mixture-UCB-CAB, and another new algorithm presented in the next section.

Mixture-UCB-CAB can be extended to the Sparse-Mixture-UCB-CAB algorithm which eventually select only a small number of models. This can be useful if there is a subscription cost for each model. Refer to Appendix 8.3 for discussions.

## 5.2 MIXTURE UPPER CONFIDENCE BOUND – ONLINE GRADIENT DESCENT

We present an alternative to Mixture-UCB-CAB, called the *mixture upper confidence bound – online gradient descent (Mixture-UCB-OGD) algorithm*, inspired by the online gradient descent algorithm (Shalev-Shwartz et al., 2012). It also has a parameter $\beta > 1$. Refer to Algorithm 2.

Mixture-UCB-CAB and Mixture-UCB-OGD can both be regarded as generalizations of the original UCB algorithm, in the sense that they reduce to UCB when $\kappa(x, x') = 0$. If we remove the $\frac{2}{t}\hat{\mathbf{K}}(\mathbf{x})\mathbf{n}^{(t)}$ term in (8), then Mixture-UCB-OGD becomes the same as UCB.

---

[6]Theorem 1 holds for a fixed $\boldsymbol{\alpha}$. A worst-case bound that simultaneously holds for every $\boldsymbol{\alpha}$ is in Lemma 1.

[7]To justify calling $R := \mathbb{E}[L(\hat{P}^{(T)})] - \min_{\boldsymbol{\alpha}} L(\boldsymbol{\alpha})$ the average regret per round, note that when $\kappa(x, x') = 0$ and $f(x) = -r(x)$ where $r(x)$ is the reward of the sample $x$, i.e., the loss $L(P) = \mathbb{E}_{X \sim P}[f(X)]$ is linear, $T(\mathbb{E}[L(\hat{P}^{(T)})] - \min_{\boldsymbol{\alpha}} L(\boldsymbol{\alpha})) = T \max_{i \in [m]} \mathbb{E}_{X \sim P_i}[r(X)] - \mathbb{E}[\sum_{t=1}^T r(x^{(t)})]$ indeed reduces to the conventional notion of regret. So $R$ can be regarded as the quadratic generalization of regret.

[7]We may also consider the scenario where each pull gives a batch of $l$ samples instead of only one sample. In this case, we will have $x_{b,n_b^{(t-1)}+1}, \ldots, x_{b,n_b^{(t-1)}+l} \sim P_b$ and $n_b^{(t)} = n_b^{(t-1)} + l$.

---

**Algorithm 2** Mixture-UCB-OGD

1: **Input:** $m$ generative arms, number of rounds $T$
2: **Output:** Gathered samples $\mathbf{x}^{(T)}$
3: **for** $t \in \{0, \ldots, m-1\}$ **do**
4:     Pull arm $t+1$ at time $t+1$ to obtain sample $x_{t+1,1} \sim P_{t+1}$
5: **end for**
6: **for** $t \in \{m, \ldots, T-1\}$ **do**
7:     Compute the gradient

$$\mathbf{h}^{(t)} := \nabla_{\boldsymbol{\alpha}} \left( \hat{L}(\boldsymbol{\alpha}; \mathbf{x}^{(t)}) - (\boldsymbol{\epsilon}^{(t)})^\top \boldsymbol{\alpha} \right) \Big|_{\boldsymbol{\alpha} = \mathbf{n}^{(t)}/t} = \frac{2}{t} \hat{\mathbf{K}}(\mathbf{x}^{(t)}) \mathbf{n}^{(t)} + \hat{\mathbf{f}}(\mathbf{x}^{(t)}) - \boldsymbol{\epsilon}^{(t)}, \quad (8)$$

    where $\mathbf{n}^{(t)} := (n_i^{(t)})_{i \in [m]} \in \mathbb{R}^m$, and $\boldsymbol{\epsilon}^{(t)}$ is defined as in Mixture-UCB-CAB
8:     Pull arm $b = b^{(t+1)} := \operatorname{argmin}_i h_i^{(t)}$ at time $t+1$ to obtain a new sample $x_{b, n_b^{(t)}+1} \sim P_b$.
9: **end for**
10: **return** samples $\mathbf{x}^{(T)}$

---

Both Mixture-UCB-CAB and Mixture-UCB-OGD attempt to make the "proportion vector" $\mathbf{n}^{(t)}/t$ (note that $n_i^{(t)}/t$ is the proportion of samples from model $i$) approach the optimal mixture $\boldsymbol{\alpha}^*$ that minimizes $\hat{L}(\boldsymbol{\alpha})$, but they do so in different manners. Mixture-UCB-CAB first computes the estimate $\boldsymbol{\alpha}^{(t)}$ after time $t$, then approaches $\boldsymbol{\alpha}^{(t)}$ by pulling an arm randomly chosen from the distribution $\boldsymbol{\alpha}^{(t)}$. Mixture-UCB-OGD estimates the gradient $\mathbf{h}^{(t)}$ of the loss function at the current proportion vector $\mathbf{n}^{(t)}/t$, and pulls an arm that results in the steepest descent of the loss.

An advantage of Mixture-UCB-OGD is that the computation of gradient (8) is significantly faster than the quadratic program (6) in Mixture-UCB-CAB. The running time complexity of Mixture-UCB-OGD is $O(T^2 + Tm^2)$.[8] Nevertheless, a regret bound for Mixture-UCB-OGD similar to Theorem 2 seems to be difficult to derive, and is left for future research.

## 6 NUMERICAL RESULTS

We experiment on various scenarios to showcase the performance of our proposed algorithms. The experiments involve the following algorithms:

- **Mixture Oracle.** In the mixture oracle algorithm (Section 5), an oracle tells us the optimal mixture $\boldsymbol{\alpha}^*$ in advance, and we pull arms randomly according to this distribution. The optimal mixture is calculated by solving the quadratic optimization in Section 4 on a large number of samples for each arm. The number of chosen samples varies based on the experiments. This is an unrealistic setting that only serves as a theoretical upper bound of the performance of any online algorithm. A realistic algorithm that performs close to the mixture oracle would be almost optimal.

- **One-Arm Oracle.** An oracle tells us the optimal single arm in advance, and we keep pulling this arm. This is an unrealistic setting. If our algorithms outperform the one-arm oracle, this will show that the advantage of pulling a mixture of arms (instead of a single arm) can be realistically achieved via online algorithms.

- **Vanilla-UCB.** A direct application of UCB mentioned near the end of Section 5.1. This serves as a baseline for the purpose of comparison.

- **Successive Halving.** The Success Halving algorithm (Karnin et al., 2013; Jamieson & Talwalkar, 2016; Chen & Ghosh, 2024) which serves as a second baseline for comparison.

- **Mixture-UCB-CAB.** The mixture upper confidence bound – continuum-armed bandit algorithm proposed in Section 5.1.

---

[8]To update $\hat{\mathbf{K}}(\mathbf{x}^{(t)})$ after a new sample $x'$ is obtained, we only need to compute $\kappa(x, x')$ for each existing sample $x$, and add their contributions to the corresponding entries in $\hat{\mathbf{K}}(\mathbf{x}^{(t)})$, requiring a computational time that is linear with the number of existing samples.

- **Mixture-UCB-OGD.** The mixture upper confidence bound – online gradient descent algorithm proposed in Section 5.2.

**Experiments Setup.** We used DINOv2-ViT-L/14 (Oquab et al., 2023) for image feature extraction, as recommended in (Stein et al., 2023), and RoBERTa (Liu et al., 2019) as the text encoder. Detailed explanation of the setup for each experiment is presented in Section 8.4.

## 6.1 OPTIMAL MIXTURE FOR DIVERSITY AND QUALITY VIA KID

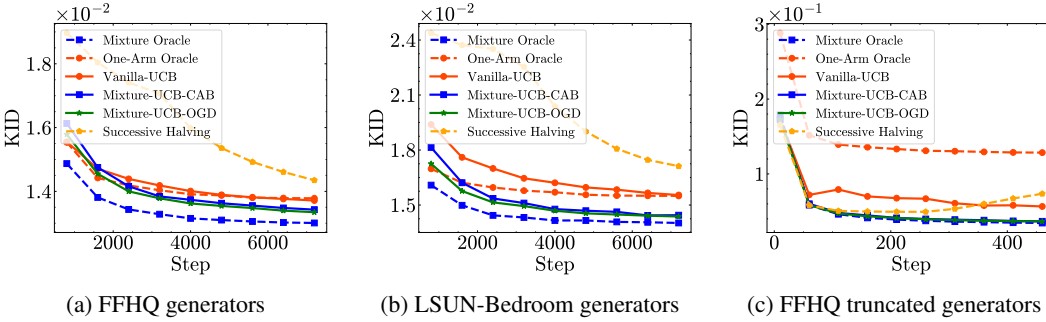

(a) FFHQ generators      (b) LSUN-Bedroom generators      (c) FFHQ truncated generators

Figure 3: Performance comparison of online algorithms for the KID metric across FFHQ, LSUN-Bedroom, and FFHQ Truncated generators.

We conducted three experiments to evaluate our method using the Kernel Inception Distance (KID) metric. In the first experiment, we used five distinct generative models: LDM (Rombach et al., 2022), StyleGAN-XL (Sauer et al., 2022), Efficient-VDVAE (Hazami et al., 2022), InsGen (Yang et al., 2021), and StyleNAT (Walton et al., 2022), all trained on the FFHQ dataset (Karras et al., 2019). In the second experiment, we used generated images from four models[9]: StyleGAN (Karras et al., 2019), Projected GAN (Sauer et al., 2021), iDDPM (Nichol & Dhariwal, 2021), and Unleashing Transformers (Bond-Taylor et al., 2022), all trained on the LSUN-Bedroom dataset (Yu et al., 2015). This experiment followed a similar setup to the first. In the final experiment, we employed the truncation method (Marchesi, 2017; Karras et al., 2019) to generate diversity-controlled images centered on eight randomly selected points, using StyleGAN2-ADA (Karras et al., 2020), also trained on the FFHQ dataset. Figure 3 demonstrates that the mixture of generators achieves better KID scores compared to individual models. Additionally, the two Mixture-UCB algorithms consistently outperform the baselines.

## 6.2 OPTIMAL MIXTURE FOR DIVERSITY VIA RKE

We used the RKE Mode Count (Jalali et al., 2023) as an evaluation metric to show the effect of mixing the models on the diversity and the advantage of our algorithms Mixture-UCB-CAB and Mixture-UCB-OGD. The score in the plots is the RKE Mode Count, written as RKE for brevity.

**Synthetic Unconditional Generative Models** We conduct two experiments on diversity-limited generative models. First, we used eight center points with a truncation value of 0.3 to generate images using StyleGAN2-ADA, trained on the FFHQ dataset. In the second experiment, we applied the same model, trained on the AFHQ Cat dataset (Choi et al., 2020), with a truncation value of 0.4. As shown in Figure 7, the optimal mixture and our algorithms consistently achieve higher RKE scores. The increase in diversity is visually depicted in Figures 6 and 8.

**Text to Image Generative Models** We used Stable Diffusion XL (Podell et al., 2024) with specific prompts to create three car image generators with distinct styles: realistic, surreal, and cartoon. In the second experiment, recognizing the importance of diversity in generative models for design tasks, we used five models—FLUX.1-Schnell (Labs, 2024), Kandinsky 3.0 (Vladimir et al., 2024), PixArt-$\alpha$ (Chen et al., 2023a), and Stable Diffusion XL—to generate images of the object "Sofa".

---

[9]FFHQ and LSUN-Bedroom datasets were downloaded from the dgm-eval repository (Stein et al., 2023) (licensed under MIT license): https://github.com/layer6ai-labs/dgm-eval.

In a similar manner, we generated red bird images using Kandinsky 3.0, Stable Diffusion 3 (Esser et al., 2024), and PixArt-$\alpha$, as shown in Figure 2.. Finally, in the third experiment, we used Stable Diffusion XL to simulate models generating images of different dog breeds: Bulldog, German Shepherd, and Poodle, respectively. This illustrates the challenge of generating diverse object types with text-to-image models. Figure 10 demonstrates the impact of using a mixture of models in the first and third experiments. The improvement in diversity is evident visually and quantitatively, as shown by the RKE scores. Our online algorithms consistently generate more diverse samples than others, as illustrated in Figure 4.

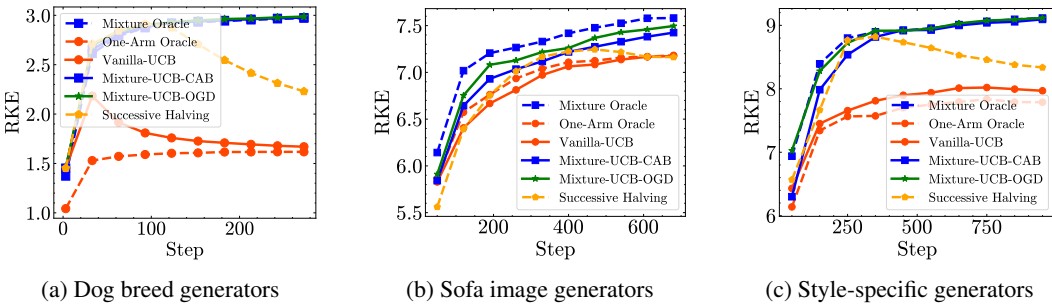

(a) Dog breed generators     (b) Sofa image generators     (c) Style-specific generators

Figure 4: Performance comparison of online algorithms using RKE score of T2I generative models.

### 6.3 OPTIMAL MIXTURE FOR DIVERSITY AND QUALITY VIA RKE AND PRECISION/DENSITY

Using RKE, we focus solely on the diversity of the arms without accounting for their quality. To address this, we apply our methodology to both RKE and Precision (Kynkäänniemi et al., 2019), as well as RKE and Density (Naeem et al., 2020). We conduct experiments in which quality is a key consideration. We use four arms: three are StyleGAN2-ADA models trained on the FFHQ dataset, each generating images with a truncation of 0.3 around randomly selected center points. The fourth model is StyleGAN2-ADA trained on CIFAR-10 (Krizhevsky et al., 2009). The FFHQ dataset is used as the reference dataset. Figures 5 and 13 demonstrate the ability of our algorithms in finding optimal mixtures with higher diversity/quality score.

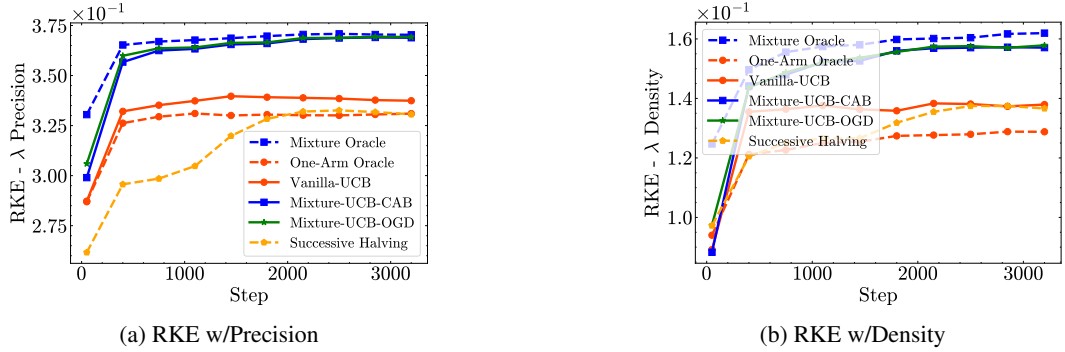

(a) RKE w/Precision                (b) RKE w/Density

Figure 5: Performance comparison of online algorithms using the combination of RKE with Precision and RKE with Density metrics.

## 7 CONCLUSION AND LIMITATIONS

We studied the online selection from several generative models, where the online learner aims to generate samples with the best overall quality and diversity. While standard multi-armed bandit (MAB) algorithms converge to one arm and select one generative model, we highlighted that a mixture of generative models could achieve a higher score compared to the individual models. We proposed the Mixture-UCB MAB algorithm to find the optimal mixture. Our experiments suggest the usefulness of the algorithm in improving the performance scores over individual arms. However, we note that the diversity gain offered by the mixture approach should be analyzed together with the quality of generated data, which can be adjusted by setting the coefficient of the quality Precision or Density score when applying Mixture-UCB. The analysis of the regret of Mixture-UCB-OGD and conditions under which a mixture can or cannot improve the scores will remain for future studies.

ACKNOWLEDGMENTS

The work of Cheuk Ting Li is partially supported by two grants from the Research Grants Council of the Hong Kong Special Administrative Region, China [Project No.s: CUHK 24205621 (ECS), CUHK 14209823 (GRF)]. The work of Farzan Farnia is partially supported by a grant from the Research Grants Council of the Hong Kong Special Administrative Region, China, Project 14209920, and is partially supported by CUHK Direct Research Grants with CUHK Project No. 4055164 and 4937054. Finally, the authors would like to thank the anonymous reviewers for their constructive feedback and suggestions.

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

# 8 APPENDIX

## 8.1 PROOF OF THEOREM 1

Consider any $\tilde{x}_{i,a} \in \mathcal{X}$. Let $\tilde{\mathbf{x}}$ be the samples which are identical to $\mathbf{x}$ except that one entry $x_{i,a}$ is changed to $\tilde{x}_{i,a}$. We have

$$
\left| \hat{L}(\boldsymbol{\alpha}; \tilde{\mathbf{x}}) - \hat{L}(\boldsymbol{\alpha}; \mathbf{x}) \right|
$$

$$
= \left| \frac{\alpha_i}{n_i} \left( f(\tilde{x}_{i,a}) - f(x_{i,a}) \right) + 2 \sum_{(j,b) \neq (i,a)} \frac{\alpha_i \alpha_j}{n_i n_j} \left( \kappa(\tilde{x}_{i,a}, x_{j,b}) - \kappa(x_{i,a}, x_{j,b}) \right) \right.
$$

$$
\left. + \frac{\alpha_i^2}{n_i^2} \left( \kappa(\tilde{x}_{i,a}, \tilde{x}_{i,a}) - \kappa(x_{i,a}, x_{i,a}) \right) \right|
$$

$$
\leq \frac{\alpha_i}{n_i} \Delta_f + 2 \sum_{(j,b) \neq (i,a)} \frac{\alpha_i \alpha_j}{n_i n_j} \Delta_\kappa + \frac{\alpha_i^2}{n_i^2} \Delta_\kappa
$$

$$
\leq \frac{\alpha_i}{n_i} (\Delta_f + 2\Delta_\kappa)
$$

$$
= \frac{\alpha_i}{n_i} \Delta_L.
$$

By McDiarmid's inequality,

$$
\mathbb{P} \left( \hat{L}(\boldsymbol{\alpha}; \mathbf{x}) - \mathbb{E}[\hat{L}(\boldsymbol{\alpha}; \mathbf{x})] \geq \epsilon \right) \leq \exp \left( - \frac{2\epsilon^2}{\sum_{i=1}^m \sum_{a=1}^{n_i} (\frac{\alpha_i}{n_i} \Delta_L)^2} \right)
$$

$$
= \exp \left( - \frac{2\epsilon^2}{\Delta_L^2 \sum_{i=1}^m \alpha_i^2 / n_i} \right).
$$

Note that

$$
\left| L(\boldsymbol{\alpha}) - \mathbb{E}[\hat{L}(\boldsymbol{\alpha}; \mathbf{x})] \right|
$$

$$
= \left| \mathbb{E}_{X, X' \overset{\text{iid}}{\sim} P}[\kappa(X, X')] - \mathbb{E} \left[ \sum_{(i,j) \in [m]^2} \frac{\alpha_i \alpha_j}{n_i n_j} \sum_{a=1}^{n_i} \sum_{b=1}^{n_j} \kappa(x_{i,a}, x_{j,b}) \right] \right|
$$

$$
= \left| \sum_{i=1}^m \frac{\alpha_i^2}{n_i^2} \sum_{a=1}^{n_i} \left( \mathbb{E}_{X, X' \overset{\text{iid}}{\sim} P}[\kappa(X, X')] - \mathbb{E}[\kappa(x_{i,a}, x_{i,a})] \right) \right|
$$

$$
\leq \sum_{i=1}^m \frac{\alpha_i^2}{n_i} \Delta_\kappa.
$$

Hence, for $\delta > 0$,

$$
\mathbb{P} \left( \hat{L}(\boldsymbol{\alpha}; \mathbf{x}) - L(\boldsymbol{\alpha}) \geq \Delta_L \sqrt{\frac{\log(1/\delta)}{2} \sum_{i=1}^m \frac{\alpha_i^2}{n_i}} + \Delta_\kappa \sum_{i=1}^m \frac{\alpha_i^2}{n_i} \right) \leq \delta.
$$

The result follows from

$$
\Delta_L \sqrt{\frac{\log(1/\delta)}{2} \sum_{i=1}^m \frac{\alpha_i^2}{n_i}} + \Delta_\kappa \sum_{i=1}^m \frac{\alpha_i^2}{n_i}
$$

$$
\leq \Delta_L \sum_{i=1}^m \sqrt{\frac{\log(1/\delta)}{2} \frac{\alpha_i^2}{n_i}} + \Delta_\kappa \sum_{i=1}^m \frac{\alpha_i}{n_i}
$$

$$
= \sum_{i=1}^m \left( \Delta_L \sqrt{\frac{\log(1/\delta)}{2 n_i}} + \frac{\Delta_\kappa}{n_i} \right) \alpha_i.
$$

The other direction of the inequality is similar.

## 8.2 PROOF OF THEOREM 2

Before we prove Theorem 2, we first prove a worst case concentration bound.

**Lemma 1** *Let $T \geq 2$, and $x_{i,1}, x_{i,2}, \ldots \overset{iid}{\sim} P_i$ for $i \in [m]$. Let $\Delta_L := 2\Delta_\kappa + \Delta_f$. For $n_1, \ldots, n_m \in [T]$, let $\mathbf{x}^{(n_i)_i} := (x_{i,a})_{i \in [m],\, a \in [n_i]}$. Fix any $\delta > 0$. With probability at least $1 - \delta$, we have*

$$L(\boldsymbol{\alpha}) - \hat{L}(\boldsymbol{\alpha}; \mathbf{x}^{(n_i)_i})$$

$$\leq \sum_{i=1}^m \left( \Delta_L \sqrt{\frac{1}{2n_i} \log \frac{m^2 T^2}{2\delta}} + \frac{\Delta_\kappa}{n_i} \right) \alpha_i.$$

*for every $n_1, \ldots, n_m \in [T]$ and probability vector $\boldsymbol{\alpha}$. The same holds for $\hat{L}(\boldsymbol{\alpha}; \mathbf{x}^{(n_i)_i}) - L(\boldsymbol{\alpha})$ instead of $L(\boldsymbol{\alpha}) - \hat{L}(\boldsymbol{\alpha}; \mathbf{x}^{(n_i)_i})$.*

*Proof:* Fix any $n_1, \ldots, n_m \in [T]$, and write $\mathbf{x} = \mathbf{x}^{(n_i)_i}$. We have

$$L(\boldsymbol{\alpha}) - \hat{L}(\boldsymbol{\alpha}; \mathbf{x})$$

$$= \sum_{i,j} \alpha_i \alpha_j \left( \frac{\mathbf{f}_i + \mathbf{f}_j}{2} + \mathbf{K}_{i,j} - \frac{\hat{\mathbf{f}}_i(\mathbf{x}) + \hat{\mathbf{f}}_j(\mathbf{x})}{2} - \hat{\mathbf{K}}_{i,j}(\mathbf{x}) \right).$$

For $i = j$, applying Theorem 1 on $\boldsymbol{\alpha}$ being the $i$-th basis vector,

$$\mathbb{P}\left( \mathbf{f}_i + \mathbf{K}_{i,i} - \hat{\mathbf{f}}_i(\mathbf{x}) - \hat{\mathbf{K}}_{i,i}(\mathbf{x}) \geq \Delta_L \sqrt{\frac{\log(1/\delta)}{2n_i}} + \frac{\Delta_\kappa}{n_i} \right) \leq \delta. \tag{9}$$

For $i \neq j$, we will use similar arguments as Theorem 1. Let $\tilde{\mathbf{x}}$ be the samples which are identical to $\mathbf{x}$ except that one entry $x_{i,a}$ of the $i$-th arm is changed to $\tilde{x}_{i,a}$. We have

$$\left| \frac{\hat{\mathbf{f}}_i(\tilde{\mathbf{x}}) + \hat{\mathbf{f}}_j(\tilde{\mathbf{x}})}{2} + \hat{\mathbf{K}}_{i,j}(\tilde{\mathbf{x}}) - \frac{\hat{\mathbf{f}}_i(\mathbf{x}) + \hat{\mathbf{f}}_j(\mathbf{x})}{2} - \hat{\mathbf{K}}_{i,j}(\mathbf{x}) \right|$$

$$= \left| \frac{1}{2n_i} \left( f(\tilde{x}_{i,a}) - f(x_{i,a}) \right) + \frac{1}{n_i n_j} \sum_{b=1}^{n_j} \left( \kappa(\tilde{x}_{i,a}, x_{j,b}) - \kappa(x_{i,a}, x_{j,b}) \right) \right|$$

$$\leq \frac{\Delta_f}{2n_i} + \frac{\Delta_\kappa}{n_i}$$

$$= \frac{\Delta_L}{2n_i}.$$

Note that $\mathbb{E}\left[\frac{\hat{\mathbf{f}}_i(\mathbf{x}) + \hat{\mathbf{f}}_j(\mathbf{x})}{2} + \hat{\mathbf{K}}_{i,j}(\mathbf{x})\right] = \frac{\mathbf{f}_i + \mathbf{f}_j}{2} - \mathbf{K}_{i,j}$. By McDiarmid's inequality,

$$\mathbb{P}\left( \frac{\mathbf{f}_i + \mathbf{f}_j}{2} + \mathbf{K}_{i,j} - \frac{\hat{\mathbf{f}}_i(\mathbf{x}) + \hat{\mathbf{f}}_j(\mathbf{x})}{2} - \hat{\mathbf{K}}_{i,j}(\mathbf{x}) \geq \epsilon \right)$$

$$\leq \exp\left( -\frac{2\epsilon^2}{n_i \left(\frac{\Delta_L}{2n_i}\right)^2 + n_j \left(\frac{\Delta_L}{2n_j}\right)^2} \right)$$

$$= \exp\left( -\frac{8\epsilon^2}{\Delta_L^2 \left(n_i^{-1} + n_j^{-1}\right)} \right).$$

Hence,

$$\mathbb{P}\left( \frac{\mathbf{f}_i + \mathbf{f}_j}{2} + \mathbf{K}_{i,j} - \frac{\hat{\mathbf{f}}_i(\mathbf{x}) + \hat{\mathbf{f}}_j(\mathbf{x})}{2} - \hat{\mathbf{K}}_{i,j}(\mathbf{x}) \geq \Delta_L \sqrt{\frac{\log(1/\delta)}{8} \left(n_i^{-1} + n_j^{-1}\right)} \right) \leq \delta. \tag{10}$$

Note that the event in (9) does not depend on $n_{i'}$ for $i' \neq i$, and the event in (10) does not depend on $n_{i'}$ for $i' \notin \{i, j\}$. By union bound, all the events in (9) and (10) do not hold for all $i \leq j$ and $n_1, \ldots, n_m \in [T]$ with probability at least

$$1 - mT\delta - \frac{m(m-1)}{2}T^2\delta \geq 1 - \frac{m^2}{2}T^2\delta.$$

If these events do not hold, then

$$
\begin{aligned}
&L(\boldsymbol{\alpha}) - \hat{L}(\boldsymbol{\alpha}; \mathbf{x}) \\
&= \sum_{i,j} \alpha_i \alpha_j \left( \frac{\mathbf{f}_i + \mathbf{f}_j}{2} + \mathbf{K}_{i,j} - \frac{\hat{\mathbf{f}}_i(\mathbf{x}) + \hat{\mathbf{f}}_j(\mathbf{x})}{2} - \hat{\mathbf{K}}_{i,j}(\mathbf{x}) \right) \\
&\leq \sum_i \alpha_i^2 \left( \Delta_L \sqrt{\frac{\log(1/\delta)}{2n_i}} + \frac{\Delta_\kappa}{n_i} \right) + \sum_{(i,j) \in [m]^2, i \neq j} \alpha_i \alpha_j \Delta_L \sqrt{\frac{\log(1/\delta)}{8}} \left( n_i^{-1} + n_j^{-1} \right) \\
&\leq \sum_i \alpha_i^2 \left( \Delta_L \sqrt{\frac{\log(1/\delta)}{2n_i}} + \frac{\Delta_\kappa}{n_i} \right) + \Delta_L \sqrt{\frac{\log(1/\delta)}{8}} \sum_{(i,j) \in [m]^2, i \neq j} \alpha_i \alpha_j \left( n_i^{-1/2} + n_j^{-1/2} \right) \\
&= \sum_i \alpha_i^2 \frac{\Delta_\kappa}{n_i} + \Delta_L \sqrt{\frac{\log(1/\delta)}{8}} \sum_{(i,j) \in [m]^2} \alpha_i \alpha_j \left( n_i^{-1/2} + n_j^{-1/2} \right) \\
&= \sum_i \alpha_i^2 \frac{\Delta_\kappa}{n_i} + \Delta_L \sqrt{\frac{\log(1/\delta)}{2}} \sum_i \alpha_i n_i^{-1/2} \\
&\leq \sum_i \left( \Delta_L \sqrt{\frac{\log(1/\delta)}{2n_i}} + \frac{\Delta_\kappa}{n_i} \right) \alpha_i.
\end{aligned}
$$

The other direction of the inequality is similar.

We finally prove Theorem 2.

*Proof:* Assume $m \geq 2$, $\beta \geq 4$ and $T \geq 2$. If $T \leq 40m$, then since $T^{-1} \log T$ is decreasing for $T \geq 3$ (the following inequalities are obviously true for $T = 2$),

$$\sqrt{\frac{\beta m \log T}{T}} \geq \sqrt{\frac{4m \log(40m)}{40m}} \geq \sqrt{\frac{\log 80}{10}} \geq 0.66,$$

and Theorem 2 is trivially true since $\mathbb{E}\left[L(\hat{P}^{(T)})\right] - \min_{\boldsymbol{\alpha}} L(\boldsymbol{\alpha}) \leq \Delta_L$. Hence we can assume $T \geq 40m + 1$.

Let $\boldsymbol{\alpha}^*$ be the minimizer of $L(\boldsymbol{\alpha})$. Let $\bar{x}^{(t)}$ be the sample obtained at the $t$-th pull. Let $\alpha_i^{(t-1)} = \mathbf{1}\{t = i\}$ for $t \in [m]$, so "$\bar{x}^{(t)}$ is generated from the distribution $P_i$ with probability $\alpha_i^{(t-1)}$" holds for every $t \geq 1$. Write $\bar{x}^{([s])} := (\bar{x}^{(t)})_{t \in [s]}$. For $s < t$, let $\hat{x}^{(s)}$ be a random variable with the same conditional distribution given $\bar{x}^{([s-1])}$ as $\bar{x}^{(s)}$, but is conditionally independent of all other random variables given $\bar{x}^{([s-1])}$. The joint distribution of $\bar{x}^{([t-1])}, \bar{x}^{(t)}, \hat{x}^{(s)}$ is

$$P_{\bar{x}^{([t-1])}, \bar{x}^{(t)}, \hat{x}^{(s)}} = P_{\bar{x}^{([t-1])}, \bar{x}^{(t)}} P_{\hat{x}^{(s)} | \bar{x}^{([s-1])}}.$$

Recall that $\bar{x}^{(s)}$ is generated from the distribution $P_i$ with probability $\alpha_i^{(s-1)}$ for $i \in [m]$, where $\boldsymbol{\alpha}^{(s-1)} = (\alpha_i^{(s-1)})_{i \in [m]}$ is computed using $\bar{x}^{([s-1])}$. We have

$$
\begin{aligned}
&\mathbb{E}[\kappa(\hat{x}^{(s)}, \bar{x}^{(t)})] \\
&= \mathbb{E}\left[ \mathbb{E}\left[ \kappa(\hat{x}^{(s)}, \bar{x}^{(t)}) \mid \bar{x}^{([t-1])} \right] \right] \\
&\overset{(a)}{=} \mathbb{E}\left[ \sum_{i=1}^m \sum_{j=1}^m \alpha_i^{(s-1)} \alpha_i^{(t-1)} \mathbb{E}_{X \sim P_i, X' \sim P_j}\left[ \kappa(X, X') \right] \right] \\
&= \mathbb{E}\left[ (\boldsymbol{\alpha}^{(s-1)})^\top \mathbf{K} \boldsymbol{\alpha}^{(t-1)} \right],
\end{aligned}
$$

where (a) is because $\hat{x}^{(s)}$ only depends on $\bar{x}^{([s-1])}$ (i.e., is conditionally independent of all other random variables in the expression given $\bar{x}^{([s-1])}$), and $\bar{x}^{(t)}$ only depends on $\bar{x}^{([t-1])}$. Write $\delta_{\mathrm{TV}}(A\|B)$ for the total variation distance between the distributions of the random variables $A$ and $B$. Write $I(A; B|C)$ for the conditional mutual information between $A$ and $B$ given $C$ in nats. We have

$$
\begin{aligned}
& \mathbb{E}[\kappa(\bar{x}^{(s)}, \bar{x}^{(t)})] \\
& \overset{(b)}{\leq} \mathbb{E}[\kappa(\hat{x}^{(s)}, \bar{x}^{(t)})] + \Delta_\kappa \delta_{\mathrm{TV}}(\bar{x}^{(s)}, \bar{x}^{(t)} \| \hat{x}^{(s)}, \bar{x}^{(t)}) \\
& \leq \mathbb{E}\left[(\boldsymbol{\alpha}^{(s-1)})^\top \mathbf{K}\boldsymbol{\alpha}^{(t-1)}\right] + \Delta_\kappa \delta_{\mathrm{TV}}(\bar{x}^{([s-1])}, \bar{x}^{(s)}, \bar{x}^{(t)} \| \bar{x}^{([s-1])}, \hat{x}^{(s)}, \bar{x}^{(t)}) \\
& \overset{(c)}{\leq} \mathbb{E}\left[(\boldsymbol{\alpha}^{(s-1)})^\top \mathbf{K}\boldsymbol{\alpha}^{(t-1)}\right] + \Delta_\kappa \sqrt{\frac{1}{2}I(\bar{x}^{(s)}; \bar{x}^{(t)}|\bar{x}^{([s-1])})},
\end{aligned}
$$

where (b) is because $\kappa$ takes values over $[\kappa_0, \kappa_1]$ with $\Delta_\kappa = \kappa_1 - \kappa_0$, and (c) is by Pinsker's inequality. We also have, for every $t$,

$$
\mathbb{E}[\kappa(\bar{x}^{(t)}, \bar{x}^{(t)})] \leq \mathbb{E}\left[(\boldsymbol{\alpha}^{(t-1)})^\top \mathbf{K}\boldsymbol{\alpha}^{(t-1)}\right] + \Delta_\kappa.
$$

Hence,

$$
\begin{aligned}
& \mathbb{E}\left[\frac{1}{T^2}\sum_{s=1}^{T}\sum_{t=1}^{T}\kappa(\bar{x}^{(s)}, \bar{x}^{(t)})\right] - \mathbb{E}\left[\frac{1}{T^2}\sum_{s=1}^{T}\sum_{t=1}^{T}(\boldsymbol{\alpha}^{(s-1)})^\top \mathbf{K}\boldsymbol{\alpha}^{(t-1)}\right] \\
& \leq \frac{2\Delta_\kappa}{T^2}\sum_{s=1}^{T}\sum_{t=s+1}^{T}\sqrt{\frac{1}{2}I(\bar{x}^{(s)}; \bar{x}^{(t)}|\bar{x}^{([s-1])})} + \frac{\Delta_\kappa}{T} \\
& = \frac{2\Delta_\kappa}{T^2}\sum_{t=1}^{T}\sum_{s=1}^{t-1}\sqrt{\frac{1}{2}I(\bar{x}^{(s)}; \bar{x}^{(t)}|\bar{x}^{([s-1])})} + \frac{\Delta_\kappa}{T} \\
& \leq \frac{2\Delta_\kappa}{T^2}\sum_{t=1}^{T}\sqrt{\frac{t-1}{2}\sum_{s=1}^{t-1}I(\bar{x}^{(s)}; \bar{x}^{(t)}|\bar{x}^{([s-1])})} + \frac{\Delta_\kappa}{T} \\
& \overset{(d)}{=} \frac{2\Delta_\kappa}{T^2}\sum_{t=1}^{T}\sqrt{\frac{t-1}{2}I(\bar{x}^{([t-1])}; \bar{x}^{(t)})} + \frac{\Delta_\kappa}{T} \\
& \overset{(e)}{\leq} \frac{2\Delta_\kappa}{T^2}\sum_{t=1}^{T}\sqrt{\frac{t-1}{2}\log m} + \frac{\Delta_\kappa}{T} \\
& \leq \frac{2\Delta_\kappa}{T^2}\sqrt{\frac{\log m}{2}}\int_0^T \sqrt{\tau}\mathrm{d}\tau + \frac{\Delta_\kappa}{T} \\
& = \frac{4\Delta_\kappa}{3}\sqrt{\frac{\log m}{2T}} + \frac{\Delta_\kappa}{T},
\end{aligned}
$$

where (d) is by the chain rule of mutual information, and (e) is because $\bar{x}^{(t)}$ only depends on $\bar{x}^{([t-1])}$ through the choice of arm $b^{(t)} \in [m]$, and hence $I(\bar{x}^{([t-1])}; \bar{x}^{(t)})$ is upper bounded by the entropy of

$b^{(t)}$, which is at most $\log m$. Also note that $\mathbb{E}[f(\bar{x}^{(t)})] = \mathbf{f}^\top \mathbb{E}[\boldsymbol{\alpha}^{(t-1)}]$. Hence,

$$
\begin{aligned}
&\mathbb{E}[L(\hat{P}^{(T)})] \\
&= \mathbb{E}\left[ \frac{1}{T} \sum_{t=1}^{T} f(\bar{x}^{(t)}) + \frac{1}{T^2} \sum_{s=1}^{T} \sum_{t=1}^{T} \kappa(\bar{x}^{(s)}, \bar{x}^{(t)}) \right] \\
&\leq \mathbb{E}\left[ \frac{1}{T} \sum_{t=1}^{T} \mathbf{f}^\top \boldsymbol{\alpha}^{(t-1)} + \frac{1}{T^2} \sum_{s=1}^{T} \sum_{t=1}^{T} (\boldsymbol{\alpha}^{(s-1)})^\top \mathbf{K} \boldsymbol{\alpha}^{(t-1)} \right] + \frac{4\Delta_\kappa}{3} \sqrt{\frac{\log m}{2T}} + \frac{\Delta_\kappa}{T} \\
&= \mathbb{E}\left[ \mathbf{f}^\top \frac{1}{T} \sum_{t=1}^{T} \boldsymbol{\alpha}^{(t-1)} + \left( \frac{1}{T} \sum_{t=1}^{T} \boldsymbol{\alpha}^{(t-1)} \right)^\top \mathbf{K} \left( \frac{1}{T} \sum_{t=1}^{T} \boldsymbol{\alpha}^{(t-1)} \right) \right] + \frac{4\Delta_\kappa}{3} \sqrt{\frac{\log m}{2T}} + \frac{\Delta_\kappa}{T} \\
&= \mathbb{E}\left[ L\left( \frac{1}{T} \sum_{t=1}^{T} \boldsymbol{\alpha}^{(t-1)} \right) \right] + \frac{4\Delta_\kappa}{3} \sqrt{\frac{\log m}{2T}} + \frac{\Delta_\kappa}{T} \\
&\overset{(f)}{\leq} \frac{1}{T} \sum_{t=1}^{T} \mathbb{E}\left[ L(\boldsymbol{\alpha}^{(t-1)}) \right] + \frac{4\Delta_\kappa}{3} \sqrt{\frac{\log m}{2T}} + \frac{\Delta_\kappa}{T},
\end{aligned}
\tag{11}
$$

where (f) is because $\mathbf{K}$ is positive semidefinite, and hence $L$ is convex. Therefore, to bound the optimality gap, we study the expected loss $\mathbb{E}\left[ L(\boldsymbol{\alpha}^{(t)}) \right]$ of the estimate of the optimal mixture distribution $\boldsymbol{\alpha}^{(t)}$.

Let $\tilde{\delta} > 0$. Let $\tilde{E}$ be the event

$$
L(\boldsymbol{\alpha}) - \hat{L}(\boldsymbol{\alpha}; \mathbf{x}^{(n_i)_i}) \leq \sum_{i=1}^{m} \left( \Delta_L \sqrt{\frac{1}{2n_i} \log \frac{m^2 T^2}{2\tilde{\delta}}} + \frac{\Delta_\kappa}{n_i} \right) \alpha_i
$$

for every $n_1, \ldots, n_m \in [T]$ and probability vector $\boldsymbol{\alpha}$, as in Lemma 1. By Lemma 1, $\mathbb{P}(\tilde{E}) \geq 1 - \tilde{\delta}$.

Fix a time $t \in \{m, \ldots, T\}$. Let $E_t$ be the event

$$
\hat{L}(\boldsymbol{\alpha}^*; \mathbf{x}^{(n_i)_i}) - L(\boldsymbol{\alpha}^*) \leq \sum_{i=1}^{m} \left( \Delta_L \sqrt{\frac{\beta \log t}{2n_i}} + \frac{\Delta_\kappa}{n_i} \right) \alpha_i^*
$$

for every $n_1, \ldots, n_m \geq 1$ such that $\sum_i n_i = t$. Since

$$
\begin{aligned}
&\sum_{i=1}^{m} \left( \Delta_L \sqrt{\frac{1}{2n_i} \log \frac{m^2 t^2}{2m^2 t^{-2}/2}} + \frac{\Delta_\kappa}{n_i} \right) \alpha_i^* \\
&= \sum_{i=1}^{m} \left( \Delta_L \sqrt{\frac{1}{2n_i} \log t^4} + \frac{\Delta_\kappa}{n_i} \right) \alpha_i^* \\
&\leq \sum_{i=1}^{m} \left( \Delta_L \sqrt{\frac{\beta \log t}{2n_i}} + \frac{\Delta_\kappa}{n_i} \right) \alpha_i^*
\end{aligned}
$$

by $\beta \geq 4$, applying Lemma 1,
$$
\mathbb{P}(E_t) \geq 1 - m^2 t^{-2}/2.
\tag{12}
$$

If the event $E_t$ holds, by taking $n_i = n_i^{(t)}$,
$$
\hat{L}(\boldsymbol{\alpha}^*; \mathbf{x}^{(t)}) - L(\boldsymbol{\alpha}^*) \leq (\boldsymbol{\epsilon}^{(t)})^\top \boldsymbol{\alpha}^*.
\tag{13}
$$

If the event $\tilde{E}$ holds,
$$
\begin{aligned}
&L(\boldsymbol{\alpha}) - \hat{L}(\boldsymbol{\alpha}; \mathbf{x}^{(t)}) \\
&\leq \sum_{i=1}^{m} \left( \Delta_L \sqrt{\frac{1}{2n_i^{(t)}} \log \frac{m^2 T^2}{2\tilde{\delta}}} + \frac{\Delta_\kappa}{n_i^{(t)}} \right) \alpha_i
\end{aligned}
\tag{14}
$$

for every $\boldsymbol{\alpha}$. Combining (13) and (14) (with $\boldsymbol{\alpha} = \boldsymbol{\alpha}^{(t)}$),

$$\hat{L}(\boldsymbol{\alpha}^{(t)}; \mathbf{x}^{(t)}) - \hat{L}(\boldsymbol{\alpha}^*; \mathbf{x}^{(t)}) + (\boldsymbol{\epsilon}^{(t)})^\top \boldsymbol{\alpha}^*$$
$$+ \sum_{i=1}^{m} \left( \Delta_L \sqrt{\frac{1}{2n_i^{(t)}} \log \frac{m^2 T^2}{2\tilde{\delta}}} + \frac{\Delta_\kappa}{n_i^{(t)}} \right) \alpha_i^{(t)}$$
$$\geq L(\boldsymbol{\alpha}^{(t)}) - L(\boldsymbol{\alpha}^*).$$

By (6), $\hat{L}(\boldsymbol{\alpha}^{(t)}; \mathbf{x}^{(t)}) - (\boldsymbol{\epsilon}^{(t)})^\top \boldsymbol{\alpha}^{(t)} \leq \hat{L}(\boldsymbol{\alpha}^*; \mathbf{x}^{(t)}) - (\boldsymbol{\epsilon}^{(t)})^\top \boldsymbol{\alpha}^*$, and hence if the events $\tilde{E}, E_t$ hold,

$$(\boldsymbol{\epsilon}^{(t)})^\top \boldsymbol{\alpha}^{(t)} + \sum_{i=1}^{m} \left( \Delta_L \sqrt{\frac{1}{2n_i^{(t)}} \log \frac{m^2 T^2}{2\tilde{\delta}}} + \frac{\Delta_\kappa}{n_i^{(t)}} \right) \alpha_i^{(t)}$$
$$\geq L(\boldsymbol{\alpha}^{(t)}) - L(\boldsymbol{\alpha}^*). \tag{15}$$

We have

$$(\boldsymbol{\epsilon}^{(t)})^\top \boldsymbol{\alpha}^{(t)} + \sum_{i=1}^{m} \left( \Delta_L \sqrt{\frac{1}{2n_i^{(t)}} \log \frac{m^2 T^2}{2\tilde{\delta}}} + \frac{\Delta_\kappa}{n_i^{(t)}} \right) \alpha_i^{(t)}$$
$$= \sum_{i=1}^{m} \left( \Delta_L \sqrt{\frac{\beta \log t}{2n_i^{(t)}}} + \frac{\Delta_\kappa}{n_i^{(t)}} + \Delta_L \sqrt{\frac{1}{2n_i^{(t)}} \log \frac{m^2 T^2}{2\tilde{\delta}}} + \frac{\Delta_\kappa}{n_i^{(t)}} \right) \alpha_i^{(t)}$$
$$= \sum_i \left( \frac{\Delta_L}{\sqrt{n_i^{(t)}}} \left( \sqrt{\frac{\beta}{2} \log t} + \sqrt{\frac{1}{2} \log \frac{m^2 T^2}{2\tilde{\delta}}} \right) + \frac{2\Delta_\kappa}{n_i^{(t)}} \right) \alpha_i^{(t)}$$
$$\leq \sum_i \left( \frac{\Delta_L}{\sqrt{n_i^{(t)}}} \left( \sqrt{\frac{\beta}{2} \log T} + \sqrt{\frac{1}{2} \log \frac{m^2 T^2}{2\tilde{\delta}}} \right) + \frac{\Delta_L}{\sqrt{n_i^{(t)}}} \right) \alpha_i^{(t)}$$
$$= \Delta_L \eta \sum_i \frac{\alpha_i^{(t)}}{\sqrt{n_i^{(t)}}},$$

where

$$\eta := \sqrt{\frac{\beta}{2} \log T} + \sqrt{\frac{1}{2} \log \frac{m^2 T^2}{2\tilde{\delta}}} + 1.$$

Substituting into (15), if the events $\tilde{E}, E_t$ hold,

$$\Delta_L \eta \sum_i \frac{\alpha_i^{(t)}}{\sqrt{n_i^{(t)}}} \geq L(\boldsymbol{\alpha}^{(t)}) - L(\boldsymbol{\alpha}^*).$$

Hence, in general (regardless of whether $\tilde{E}, E_t$ hold), denoting the indicator function of $\tilde{E} \cap E_t$ as $\mathbf{1}_{\tilde{E} \cap E_t} \in \{0, 1\}$,

$$\sum_i \frac{\alpha_i^{(t)}}{\sqrt{n_i^{(t)}}} \geq \frac{L(\boldsymbol{\alpha}^{(t)}) - L(\boldsymbol{\alpha}^*)}{\Delta_L \eta} \mathbf{1}_{\tilde{E} \cap E_t}.$$

Let

$$\Psi^{(t)} := \sum_{i=1}^{m} \psi(n_i^{(t)} - 1),$$

where $\psi(n) := \sum_{i=1}^{n} i^{-1/2}$. Recall that we pull arm $i$ at time $t+1$ with probability $\alpha_i^{(t)}$. The expected increase of $\Psi^{(t)}$ is

$$
\begin{aligned}
\mathbb{E}\left[\Psi^{(t+1)} - \Psi^{(t)} \mid \mathbf{x}^{(t)}\right] &= \sum_{i=1}^{m} \left(\psi(n_i^{(t)}) - \psi(n_i^{(t)} - 1)\right) \alpha_i^{(t)} \\
&= \sum_{i=1}^{m} \frac{\alpha_i^{(t)}}{\sqrt{n_i^{(t)}}} \\
&\geq \frac{L(\boldsymbol{\alpha}^{(t)}) - L(\boldsymbol{\alpha}^*)}{\Delta_L \eta} \mathbf{1}_{\tilde{E} \cap E_t}.
\end{aligned}
$$

Note that

$$
\begin{aligned}
\Psi^{(T)} &= \sum_{i=1}^{m} \psi(n_i^{(T)} - 1) \\
&\leq \sum_{i=1}^{m} \int_0^{n_i^{(T)} - 1} \min\{\tau^{-1/2}, 1\} \mathrm{d}\tau \\
&\overset{(a)}{\leq} m \int_0^{m^{-1} \sum_{i=1}^{m} n_i^{(T)} - 1} \min\{\tau^{-1/2}, 1\} \mathrm{d}\tau \\
&= m \int_0^{T/m - 1} \min\{\tau^{-1/2}, 1\} \mathrm{d}\tau \\
&\overset{(b)}{=} m \left(1 + \int_1^{T/m - 1} \tau^{-1/2} \mathrm{d}\tau\right) \\
&= m \left(2\sqrt{\frac{T}{m} - 1} - 1\right),
\end{aligned}
$$

where (a) is because $a \mapsto \int_0^a \min\{\tau^{-1/2}, 1\} \mathrm{d}\tau$ is concave, and (b) is because $T \geq 40m + 1$, so $T/m - 1 \geq 1$. Therefore,

$$
\begin{aligned}
m \left(2\sqrt{\frac{T}{m} - 1} - 1\right) &\geq \mathbb{E}\left[\Psi^{(T)} - \Psi^{(m)}\right] \\
&= \sum_{t=m}^{T-1} \mathbb{E}\left[\Psi^{(t+1)} - \Psi^{(t)}\right] \\
&\geq \sum_{t=m}^{T-1} \mathbb{E}\left[\frac{L(\boldsymbol{\alpha}^{(t)}) - L(\boldsymbol{\alpha}^*)}{\Delta_L \eta} \mathbf{1}_{\tilde{E} \cap E_t}\right] \\
&\geq \frac{1}{\Delta_L \eta} \sum_{t=m}^{T-1} \left(\mathbb{E}\left[L(\boldsymbol{\alpha}^{(t)}) - L(\boldsymbol{\alpha}^*)\right] - \Delta_L \mathbb{P}((\tilde{E} \cap E_t)^c)\right) \\
&\overset{(c)}{\geq} \frac{1}{\Delta_L \eta} \sum_{t=m}^{T-1} \left(\mathbb{E}\left[L(\boldsymbol{\alpha}^{(t)}) - L(\boldsymbol{\alpha}^*)\right] - \Delta_L(\tilde{\delta} + m^2 t^{-2}/2)\right) \\
&\geq \frac{1}{\Delta_L \eta} \sum_{t=m}^{T-1} \mathbb{E}\left[L(\boldsymbol{\alpha}^{(t)}) - L(\boldsymbol{\alpha}^*)\right] - \frac{T\tilde{\delta}}{\eta} - \frac{m^2}{2\eta} \int_{m-1}^{T-1} t^{-2} \mathrm{d}t \\
&\geq \frac{1}{\Delta_L \eta} \sum_{t=m}^{T-1} \mathbb{E}\left[L(\boldsymbol{\alpha}^{(t)}) - L(\boldsymbol{\alpha}^*)\right] - \frac{T\tilde{\delta}}{\eta} - \frac{m^2}{2\eta(m-1)} \\
&\geq \frac{1}{\Delta_L \eta} \sum_{t=m}^{T-1} \mathbb{E}\left[L(\boldsymbol{\alpha}^{(t)}) - L(\boldsymbol{\alpha}^*)\right] - \frac{T\tilde{\delta}}{\eta} - \frac{m}{\eta},
\end{aligned}
$$

where (c) is by (12). Hence,

$$\frac{1}{\Delta_L} \sum_{t=0}^{T-1} \mathbb{E}\left[L(\boldsymbol{\alpha}^{(t)}) - L(\boldsymbol{\alpha}^*)\right]$$

$$\leq \frac{1}{\Delta_L} \sum_{t=m}^{T-1} \mathbb{E}\left[L(\boldsymbol{\alpha}^{(t)}) - L(\boldsymbol{\alpha}^*)\right] + m$$

$$\leq \eta m \left(2\sqrt{\frac{T}{m}-1}-1\right) + T\tilde{\delta} + 2m$$

$$\overset{(d)}{=} m\left(2\sqrt{\frac{T}{m}-1}-1\right)\left(\sqrt{\frac{\beta}{2}\log T} + \sqrt{\frac{1}{2}\log\frac{mT^3}{2}}+1\right) + 3m$$

$$\overset{(e)}{\leq} m\left(2\sqrt{\frac{T}{m}}-1\right)\left(\sqrt{\frac{\beta}{2}\log T} + \sqrt{\frac{1}{2}\log\frac{mT^3}{2}}+1\right)$$

$$\leq 2\sqrt{mT}\left(\sqrt{\frac{\beta}{2}\log T} + \sqrt{\frac{1}{2}\log\frac{mT^3}{2}}+1\right),$$

where (d) is by substituting $\tilde{\delta} = m/T$, (e) is because $\sqrt{\frac{\beta}{2}\log T} \geq \sqrt{2\log 81} \geq 2.9$ and $\sqrt{\frac{1}{2}\log\frac{mT^3}{2}} \geq 2.5$ (recall that $T \geq 40m+1 \geq 81$). Substituting into (11),

$$\frac{1}{\Delta_L}\left(\mathbb{E}[L(\hat{P}^{(T)})] - L(\boldsymbol{\alpha}^*)\right)$$

$$\leq \frac{1}{\Delta_L T}\sum_{t=1}^{T}\mathbb{E}\left[L(\boldsymbol{\alpha}^{(t-1)}) - L(\boldsymbol{\alpha}^*)\right] + \frac{4\Delta_\kappa}{3\Delta_L}\sqrt{\frac{\log m}{2T}} + \frac{\Delta_\kappa}{T\Delta_L}$$

$$\leq 2\sqrt{\frac{m}{T}}\left(\sqrt{\frac{\beta}{2}\log T} + \sqrt{\frac{1}{2}\log\frac{mT^3}{2}}+1\right) + \frac{2}{3}\sqrt{\frac{\log m}{2T}} + \frac{1}{2T}$$

$$= \frac{1}{\sqrt{T}}\left(2\sqrt{\frac{\beta}{2}m\log T} + \sqrt{2m\log\frac{mT^3}{2}} + 2\sqrt{m} + \frac{2}{3}\sqrt{\frac{\log m}{2}} + \frac{1}{2\sqrt{T}}\right)$$

$$\leq \frac{1}{\sqrt{T}}\left(2\sqrt{\frac{\beta}{2}m\log T} + \sqrt{2m\log T^4} + \frac{2\sqrt{m\log T}}{\sqrt{\log 81}}\right.$$

$$\left. + \frac{2}{3}\frac{\sqrt{m\log T}}{\sqrt{m\log(40m+1)}}\sqrt{\frac{\log m}{2}} + \frac{1}{2\sqrt{81}}\cdot\frac{\sqrt{m\log T}}{\sqrt{2\log 81}}\right)$$

$$= \sqrt{\frac{m\log T}{T}}\left(\sqrt{2\beta} + 2\sqrt{2} + \frac{2}{\sqrt{\log 81}} + \frac{2}{3}\sqrt{\frac{\log m}{2m\log(40m+1)}} + \frac{1}{2\sqrt{81}}\cdot\frac{1}{\sqrt{2\log 81}}\right)$$

$$\leq \sqrt{\frac{m\log T}{T}}\left(\sqrt{2\beta} + 2\sqrt{2} + \frac{2}{\sqrt{\log 81}} + \frac{2}{3}\sqrt{\frac{\log 2}{4\log 81}} + \frac{1}{2\sqrt{81}}\cdot\frac{1}{\sqrt{2\log 81}}\right)$$

$$\leq \sqrt{\frac{m\log T}{T}}\left(\sqrt{2\beta} + 3.934\right)$$

$$\leq \sqrt{\frac{m\log T}{T}}\left(\sqrt{2\beta} + \frac{3.934}{2}\sqrt{\beta}\right)$$

$$\leq 3.382\sqrt{\frac{\beta m\log T}{T}}.$$

This completes the proof of Theorem 2 (with an improved constant 3.382 instead of 4).

### 8.3 SPARSE MIXTURE UPPER CONFIDENCE BOUND-CONTINUUM-ARMED BANDIT ALGORITHM

The optimal mixture may involve a large number of models. It is sometimes of interest to identify a small subset of models that can still give diverse samples. We now consider the scenario where there is a cost associated with each arm. If we have to pay a cost per pull, then this cost can be absorbed into the function $f$, and the problem reduces to the aforementioned quadratic multi-armed bandit. However, if the cost is a "subscription fee" that we have to pay for each arm at each round, even if we do not pull that arm at that time, until we decide to "unsubscribe" the arm and not pull it anymore, then we have to modify the algorithm to minimize the following average cost

$$L(\hat{P}^{(T)}) + \frac{\lambda}{T} \sum_{i=1}^{m} \max\{t \in [T] : b^{(t)} = i\}, \tag{16}$$

where $\hat{P}^{(T)}$ is the empirical distribution of the first $T$ samples, $b^{(t)}$ is the arm pulled at time $t$, $\max\{t \in [T] : b^{(t)} = i\}$ is the last time we pull arm $i$, and $\lambda$ is the subscription fee per round. Intuitively, we have to subscribe to arm $i$ until the last use time $\max\{t \in [T] : b^{(t)} = i\}$. As $T \to \infty$, we hope that the average cost (16) approaches the optimal cost $\min_{\boldsymbol{\alpha}} (L(\boldsymbol{\alpha}) + \lambda\|\boldsymbol{\alpha}\|_0)$, where $\|\boldsymbol{\alpha}\|_0$ is the number of nonzero entries of $\boldsymbol{\alpha}$. Minimizing (16) allows us to simultaneously select the best subset of arms and the optimal mixture in the long run, akin to variable selection methods in statistical learning.

We now generalize the Mixture-UCB-CAB algorithm to the *sparse mixture upper confidence bound – continuum-armed bandit (Sparse-Mixture-UCB-CAB) algorithm*, which has parameters $\lambda \geq 0$ and $\beta > 1$. This algorithm is inspired by the *backward elimination method* for variable selection (**?**), which starts with all variables and gradually removing variables irrelevant to our prediction. Here, we start with a set of subscribed arms $\mathcal{S}$ that contains all arms $[m]$, and gradually dropping the worst arm $i'$ as long as the upper confidence bound of the optimal cost without arm $i'$ is lower than the lower confidence bound of the optimal cost with arm $i'$, which implies that dropping arm $i'$ will have a high likelihood of reducing the cost. The algorithm is given in Algorithm 3, and the experiments are presented in Appendix 8.4.

Asymptotically, Sparse-Mixture-UCB-CAB attempts to minimize the cost $\min_{\boldsymbol{\alpha}} (L(\boldsymbol{\alpha}) + \lambda\|\boldsymbol{\alpha}\|_0)$. If a fixed sparsity $\ell$ is desired instead, we can start with $\lambda = 0$, and gradually increase $\lambda$ at each round until $|\mathcal{S}| = \ell$, and then stop unsubscribing arms.

### 8.4 DETAILS OF THE NUMERICAL EXPERIMENTS

**Hyper-parameter Choice.** The kernel bandwidths for the RKE and KID metrics were chosen based on the guidelines provided in their respective papers to ensure clear distinction between models. The values for $\Delta_L$ and $\Delta_\kappa$ in our online algorithms (7) were set according to the magnitudes of the metrics and their behavior on a validation subset. The number of sampling rounds was adjusted according to the number of arms and metric convergence, both of which depend on the bandwidth. To ensure the statistical significance of results, all experiments were repeated 10 times with different random seeds, and the reported plots represent the average results.

#### 8.4.1 OPTIMAL MIXTURE FOR QUALITY AND DIVERSITY VIA KID

Suppose $P$ is the distribution of generated images of a model, and $Q$ is the target distribution. Recall that for KID (3), we take the quadratic term to be $\kappa(x, x') = k(\psi(x), \psi(x'))$ (with an expectation $\mathbb{E}_{X,X'\sim P}[k(\psi(X), \psi(X'))])$ and the linear term to be $f(x) = -2\mathbb{E}_{Y\sim Q}[k(\psi(x), \psi(Y))]$ (with an expectation $-2\mathbb{E}_{X\sim P, Y\sim Q}[k(\psi(X), \psi(Y))]$). In order to run our online algorithms, we use $\Delta_L$ and $\Delta_\kappa$ based on a validation portion to make sure the UCB terms have the right magnitude for forcing exploration.

**FFHQ Generated Images.** In this experiment, we used images generated by five different models: LDM (Rombach et al., 2022), StyleGAN-XL (Sauer et al., 2022), Efficient-VDVAE (Hazami et al., 2022), InsGen (Yang et al., 2021), and StyleNAT (Walton et al., 2022). We used 10,000 images from each model to determine the optimal mixture. A kernel bandwidth of 40 was used for calculating

---

**Algorithm 3** Sparse-Mixture-UCB-CAB

1: **Input:** $m$ generative arms, number of rounds $T$
2: **Output:** Gathered samples $\mathbf{x}^{(T)}$
3: Initialize the set of subscribed arms $\mathcal{S} \leftarrow [m]$.
4: **for** $t \in \{0, \ldots, m-1\}$ **do**
5:     Pull arm $t+1$ at time $t+1$ to obtain sample $x_{t+1,1} \sim P_{t+1}$. Set $n_{t+1}^{(m)} = 1$.
6: **end for**
7: **for** $t \in \{m, \ldots, T-1\}$ **do**
8:     **repeat**
9:         Compute

$$\boldsymbol{\alpha}^{(t)} \leftarrow \operatorname*{argmin}_{\boldsymbol{\alpha}:\, \mathrm{supp}(\boldsymbol{\alpha}) \subseteq \mathcal{S}} \left( \hat{L}(\boldsymbol{\alpha}; \mathbf{x}^{(t)}) + \lambda|\mathcal{S}| - (\boldsymbol{\epsilon}^{(t)})^{\top} \boldsymbol{\alpha} \right), \tag{17}$$

where $\boldsymbol{\epsilon}^{(t)} \in \mathbb{R}^m$ is defined in (7). Let the minimum value above be $C$.
10:        Compute the following "worst arm" if $|\mathcal{S}| \geq 2$:

$$i' \leftarrow \operatorname*{argmin}_{i \in \mathcal{S}} \min_{\boldsymbol{\alpha}:\, \mathrm{supp}(\boldsymbol{\alpha}) \subseteq \mathcal{S} \backslash \{i\}} \left( \hat{L}(\boldsymbol{\alpha}; \mathbf{x}^{(t)}) + \lambda(|\mathcal{S}| - 1) + (\boldsymbol{\epsilon}^{(t)})^{\top} \boldsymbol{\alpha} \right).$$

Let the minimum value above be $C'$.
11:        **if** $C' \leq C$ **then**
12:            Unsubscribe arm $i'$ (i.e., $\mathcal{S} \leftarrow \mathcal{S} \backslash \{i'\}$)
13:        **end if**
14:     **until** no more arms are unsubscribed
15:     Generate the arm index $b^{(t+1)} \in [m]$ at random with $\mathbb{P}(b^{(t+1)} = i) = \alpha_i^{(t)}$.
16:     Pull arm $b = b^{(t+1)}$ at time $t+1$ to obtain a new sample $x_{b, n_b^{(t)}+1} \sim P_b$. Set $n_b^{(t+1)} = n_b^{(t)}+1$
        and $n_j^{(t+1)} = n_j^{(t)}$ for $j \neq b$.
17: **end for**
18: **return** samples $\mathbf{x}^{(T)}$

---

the RKE, and the online algorithms were run for 8,000 sampling rounds. Figure 1 presents a visual representation of the impact of our algorithm, along with the corresponding FID and KID scores.

In Tables 1 and 2, we observe that the Precision of the optimal mixture is similar to that of the maximum Precision score among individual models. On the other hand, the Recall-based diversity improved in the mixture case. However, the quality-measuring Density score slightly decreased for the selected mixture model, as Density is a linear score for quality that could be optimized by an individual model. On the other hand, the Coverage score of the mixture model was higher than each individual model.

Note that Precision and Density are scores on the average quality of samples. Intuitively, the quality score of a mixture of models is the average of the quality score of the individual models, and hence the quality score of a mixture cannot be better than the best individual model. On the other hand, Recall and Coverage measure the diversity of the samples, which can increase by considering a mixture of the models. To evaluate the net diversity-quality effect, we measured the FID score of the selected mixture and the best individual model, and the selected mixture model had a better FID score compared to the individual model with the best FID.

**LSUN-Bedroom** We used images generated by four different models: StyleGAN (Karras et al., 2019), Projected GAN (Sauer et al., 2021), iDDPM (Nichol & Dhariwal, 2021), and Unleashing Transformers (Bond-Taylor et al., 2022). We utilized 10,000 images from each model to compute the optimal mixture, resulting in weights of $(0.51, 0, 0.49, 0)$. A kernel bandwidth of 40 was applied, and the algorithm was run for 8,000 sampling steps. The quality and diversity scores for each model, including the results for the optimal mixture based on KID, are presented in Table 2.

**Truncated FFHQ.** We used StyleGAN2-ADA (Karras et al., 2020) trained on FFHQ dataset to generate images. We randomly chose 8 initial points and used the Truncation Method (Marchesi,

| Model | Precision ↑ | Recall ↑ | Density ↑ | Coverage ↑ | FID ↓ | KID ($\times 10^2$) ↓ |
|---|---|---|---|---|---|---|
| LDM | 0.856 ± 0.008 | 0.482 ± 0.008 | 0.959 ± 0.027 | 0.776 ± 0.006 | 189.876 ± 1.976 | 1.484 ± 0.019 |
| StyleGAN-XL | 0.798 ± 0.007 | 0.515 ± 0.007 | 0.726 ± 0.018 | 0.691 ± 0.009 | 186.163 ± 2.752 | 1.355 ± 0.028 |
| Efficient-VDVAE | 0.854 ± 0.011 | 0.143 ± 0.007 | 0.952 ± 0.033 | 0.545 ± 0.007 | 490.385 ± 4.377 | 5.339 ± 0.046 |
| InsGen | 0.76 ± 0.006 | 0.281 ± 0.007 | 0.716 ± 0.016 | 0.692 ± 0.005 | 278.235 ± 1.617 | 2.292 ± 0.025 |
| StyleNAT | 0.834 ± 0.008 | 0.478 ± 0.007 | 0.867 ± 0.023 | 0.775 ± 0.007 | 185.067 ± 2.123 | 1.442 ± 0.029 |
| Optimal Mixture (KID) | 0.818 ± 0.007 | 0.57 ± 0.008 | 0.816 ± 0.025 | 0.765 ± 0.007 | 168.127 ± 1.596 | 1.273 ± 0.018 |
| Mixture-UCB-CAB (KID) | 0.828 ± 0.007 | 0.571 ± 0.008 | 0.838 ± 0.016 | 0.787 ± 0.007 | 170.578 ± 2.075 | 1.342 ± 0.007 |
| Mixture-UCB-OGD (KID) | 0.827 ± 0.006 | 0.573 ± 0.005 | 0.825 ± 0.015 | 0.787 ± 0.006 | 170.113 ± 1.930 | 1.334 ± 0.007 |

Table 1: Quality and diversity scores for the FFHQ experiment, including precision, recall, density, coverage, and FID metrics (± standard deviation).

| Model | Precision ↑ | Recall ↑ | Density ↑ | Coverage ↑ | FID ↓ | KID ($\times 10^2$) ↓ |
|---|---|---|---|---|---|---|
| StyleGAN | 0.838 ± 0.008 | 0.446 ± 0.007 | 0.941 ± 0.019 | 0.821 ± 0.004 | 175.575 ± 2.055 | 1.559 ± 0.027 |
| Projected GAN | 0.749 ± 0.015 | 0.329 ± 0.008 | 0.592 ± 0.027 | 0.517 ± 0.008 | 324.066 ± 3.753 | 3.834 ± 0.053 |
| iDDPM | 0.838 ± 0.006 | 0.641 ± 0.006 | 0.660 ± 0.018 | 0.825 ± 0.006 | 154.680 ± 3.036 | 1.513 ± 0.035 |
| Unleashing Transformers | 0.786 ± 0.008 | 0.449 ± 0.006 | 0.649 ± 0.013 | 0.581 ± 0.013 | 339.982 ± 6.118 | 4.131 ± 0.051 |
| Optimal Mixture (KID) | 0.838 ± 0.006 | 0.589 ± 0.005 | 0.900 ± 0.016 | 0.833 ± 0.004 | 149.779 ± 2.238 | 1.369 ± 0.024 |
| Mixture-UCB-CAB (KID) | 0.835 ± 0.007 | 0.602 ± 0.007 | 0.894 ± 0.014 | 0.834 ± 0.005 | 151.28 ± 1.801 | 1.438 ± 0.024 |
| Mixture-UCB-OGD (KID) | 0.838 ± 0.008 | 0.599 ± 0.007 | 0.902 ± 0.023 | 0.834 ± 0.006 | 151.10 ± 1.906 | 1.434 ± 0.026 |

Table 2: Quality and diversity scores for the LSUN-Bedroom experiment, including precision, recall, density, coverage, and FID metrics (± standard deviation).

2017; Karras et al., 2019) to generate images with limited diversity around each of the chosen points. We used truncation value of 0.3 and generated 5000 images from each model to find the optimal mixture. The weights for the mixture was (0.07, 0.28, 0.10, 0.04, 0.21, 0.11, 0.12, 0.07). A kernel bandwidth of 40 was used, and 4,000 sampling steps were conducted.

### 8.4.2 OPTIMAL MIXTURE FOR DIVERSITY VIA RKE

**Truncated FFHQ.** We employed StyleGAN2-ADA (Karras et al., 2020), trained on the FFHQ dataset, to generate images. Eight initial points were randomly selected, and the Truncation Method (Marchesi, 2017; Karras et al., 2019) was applied with a truncation value of 0.3 to generate images with limited diversity around these points. For the quadratic optimization, 5,000 images were generated from each model, using a kernel bandwidth of 40 to identify the optimal mixture. In the online experiment, a new set of generated images was used, and sampling was conducted over 2,000 steps.

**Truncated AFHQ Cat.** Similar to the previous experiment, we used StyleGAN2-ADA to generate AFHQ Cat images. Four initial points were selected, and a truncation value of 0.6 was applied to simulate diversity-controlled models. For the quadratic optimization, 5,000 images were generated from each model, with sampling conducted over 1,200 steps to determine the optimal mixture.

**Style-Specific Generators.** We used Stable Diffusion XL to generate images of cars in distinct styles: realistic, surreal, and cartoon. For this experiment, we utilized 2,000 images from each model to determine the optimal mixture, which yielded weights of (0.67, 0.27, 0.06). This mixture increased the RKE value from 7.8 (the optimal value of the realistic images) to 9.2. We set the kernel bandwidth to 30 and executed the online algorithms over 1,000 sampling steps.

**Sofa Images.** We generated images of the object "Sofa" using prompts with environmental descriptions across the models FLUX.1-Schnell (Labs, 2024), Kandinsky 3.0 (Vladimir et al., 2024), PixArt-$\alpha$ (Chen et al., 2023a), and Stable Diffusion XL (Podell et al., 2024). Solving the RKE optimization with 1,000 images revealed that sampling 38% from FLUX and 62% from Kandinsky improved the RKE score from the one-arm optimum of 7.21 to 7.57. We set the kernel bandwidth to 30 and conducted the online experiment over 700 steps. We observed that Mixture-UCB-OGD achieved noticeably faster convergence to the optimal mixture RKE in this scenario.

The prompts followed the structure: "A *adjective* sofa is *verb* in a *location*," with the terms for Adjective, Action, and Location generated by GPT-4o (Achiam et al., 2023), specifically for the object "Sofa."

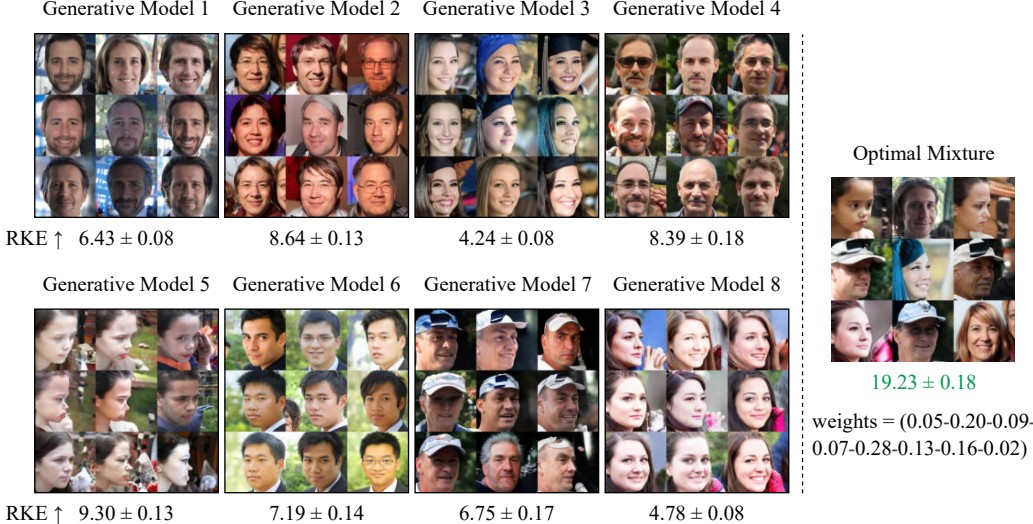

Figure 6: Visual demonstration of the increase in diversity when mixing arms compared to individual arms for truncated FFHQ generative models. The RKE values for each model and the mixture serve as indicators of diversity.

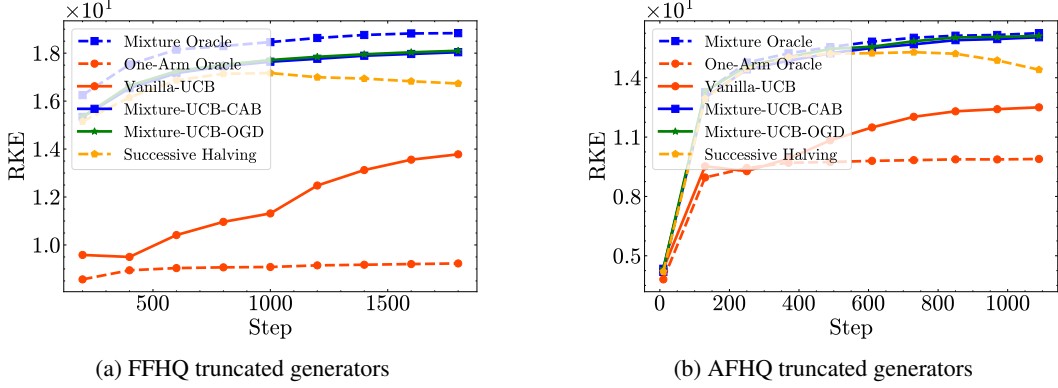

(a) FFHQ truncated generators

(b) AFHQ truncated generators

Figure 7: Performance comparison of online algorithms based on the RKE metric for Simulator Unconditional Generative Models.

**Dog Breeds Images.** Stable Diffusion XL was used to generate images of three dog breeds: Poodle, Bulldog, and German Shepherd. As shown in Figure 10, using a mixture of models resulted in an increase in mode count from 1.5 to 3, supporting our claim of enhanced diversity. We set the kernel bandwidth to 50 and generated 1,000 images for each breed to determine the optimal mixture, which was (0.33, 0.31, 0.36). Additionally, the online algorithms were executed for 500 sampling steps.

**Red Bird Images.** To observe the performance of mixing the models while generating images on a single prompt, we generate images with the prompt "Red bird, cartoon style" using Kandinsky 3.0 (Vladimir et al., 2024), Stable Diffusion 3-medium (Esser et al., 2024), and PixArt-$\alpha$ Chen et al. (2023a). We use 8000 images and a kernel bandwidth of 30 to find the optimal mixture in an offline manner. The increase in diversity is shown in Figure 2. We observe a noticeable boost in the RKE and Vendi scores, showing the diversity has improved. The performance of our online algorithms and the comparison of samples generated by each online algorithm are shown in Figure 11.

**Text Generative Models.** In this experiment, we used the OpenLLMText dataset (Chen et al., 2023b), which consists of 60,000 human texts rephrased paragraph by paragraph using the models

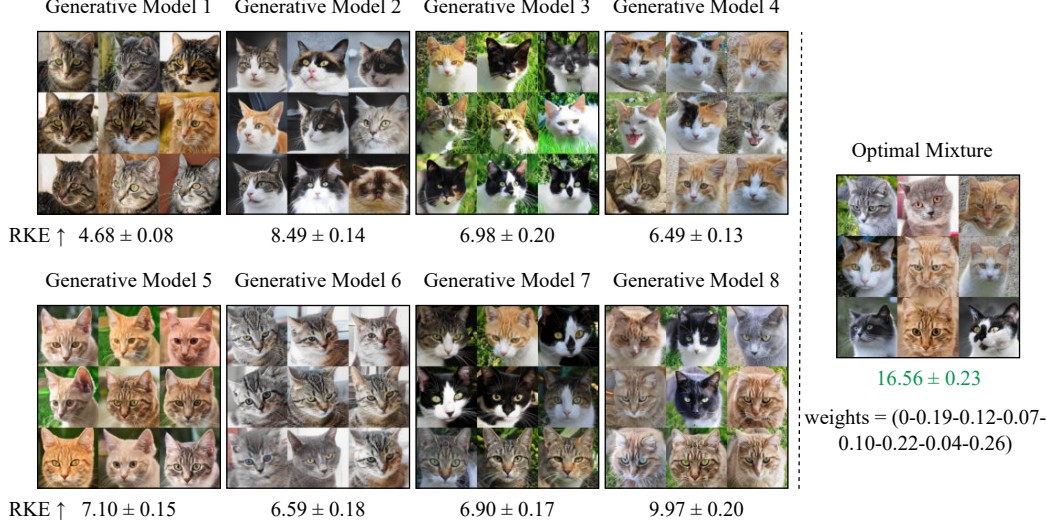

Figure 8: Visual demonstration of the increase in diversity when mixing arms compared to individual arms for truncated AFHQ Cat generative models. The RKE values for each model and the mixture represent the diversity.

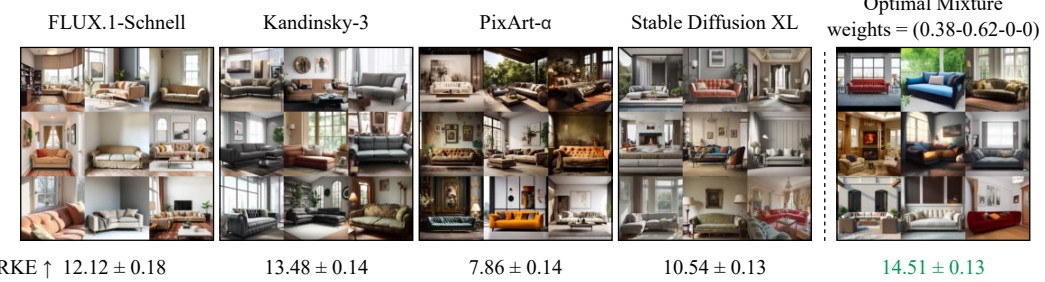

Figure 9: Visual comparison of diversity of each arm and the mixture for sofa image generators

GPT2-XL (Radford et al., 2019), LLaMA-7B (Touvron et al., 2023), and PaLM (Chowdhery et al., 2023). To extract features from each text, we employed the RoBERTa text encoder (Liu et al., 2019). By solving the optimization problem on 10,000 texts from each model, we found that mixing the models with probabilities (0.02, 0.34, 0.64, 0) achieved the optimal mixture, improving the RKE from an optimal single model score of 69.3 to 75.2. A bandwidth of 0.6 was used for the kernel, and we ran the online algorithms for 7,000 steps to demonstrate their performance. As shown in Figure 12a, the results demonstrate the advantage of our online algorithms, suggesting that our method applies not only to image generators but also to text generators.

**Sparse Mixture** Four different initial points and StyleGAN2-ADA were used to generate images with a truncation of 0.6 around the points, simulating diversity-controlled arms. A value of $\lambda = 0.06$ and a bandwidth of 30 were selected based on the magnitudes of RKEs from the validation dataset to determine when to "unsubscribe" arms in the Sparse-Mixure-UCB-CAB algorithm. We conducted three scenarios, gradually reducing the number of arms to between one and three, and presented a comparison of the resulting plots and their convergence values in Figure 12b.

### 8.4.3 OPTIMAL MIXTURE FOR DIVERSITY AND QUALITY VIA RKE AND PRECISION/DENSITY

In this experiment, we utilized four arms: three of them are StyleGAN2-ADA models trained on FFHQ, each using a truncation value of 0.3 around a randomly selected point. The fourth arm is

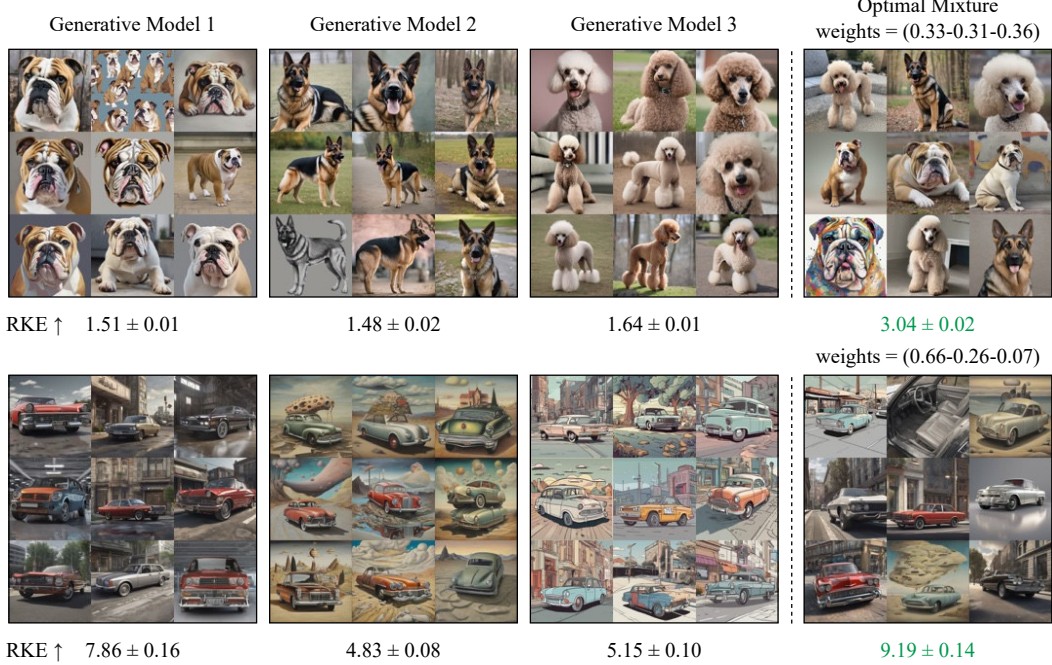

Figure 10: Visual comparison of the diversity across individual arms and the optimal mixture for Dog Breed Generators and Style-Specific Generators.

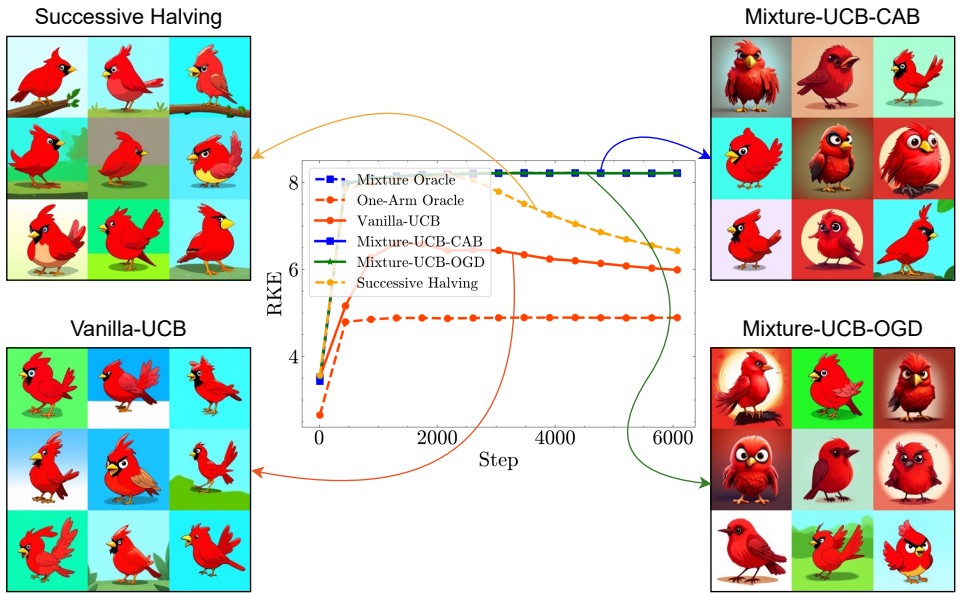

Figure 11: Comparison of our proposed algorithms Mixture-UCB-CAB and Mixture-UCB-OGD with the baseline one arm online algorithms. We plot the diversity-measuring RKE score of the generated data and display 9 (randomly-selected) samples produced in the process of each algorithm.

StyleGAN2-ADA trained on CIFAR-10. We generated 5,000 images and used a kernel bandwidth of 30 to calculate the optimal mixture. When optimizing purely for diversity using the RKE metric, the high diversity of the fourth arm leads to a probability of 0.91 being assigned to it, as shown in Figure 13. However, despite the increased diversity, the quality of the generated images, based on the reference distribution, is unsatisfactory.

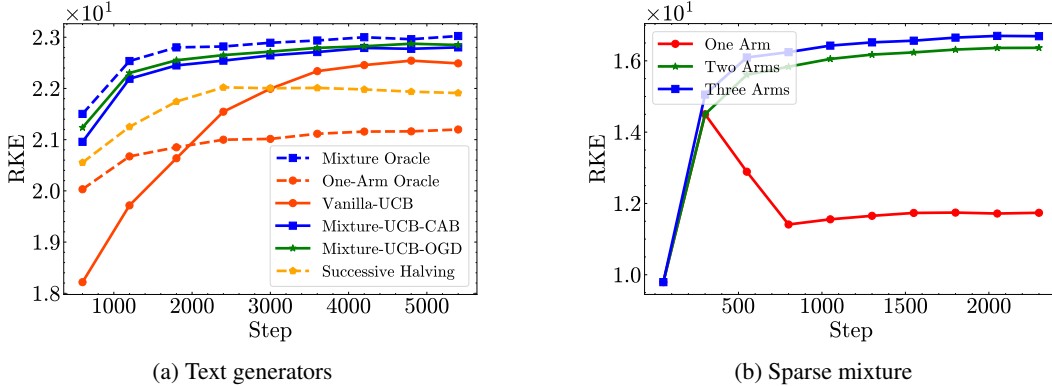

(a) Text generators

(b) Sparse mixture

Figure 12: Comparison of online algorithms for the RKE metric on text generative models and the Sparse Mixture algorithm for FFHQ truncated generators

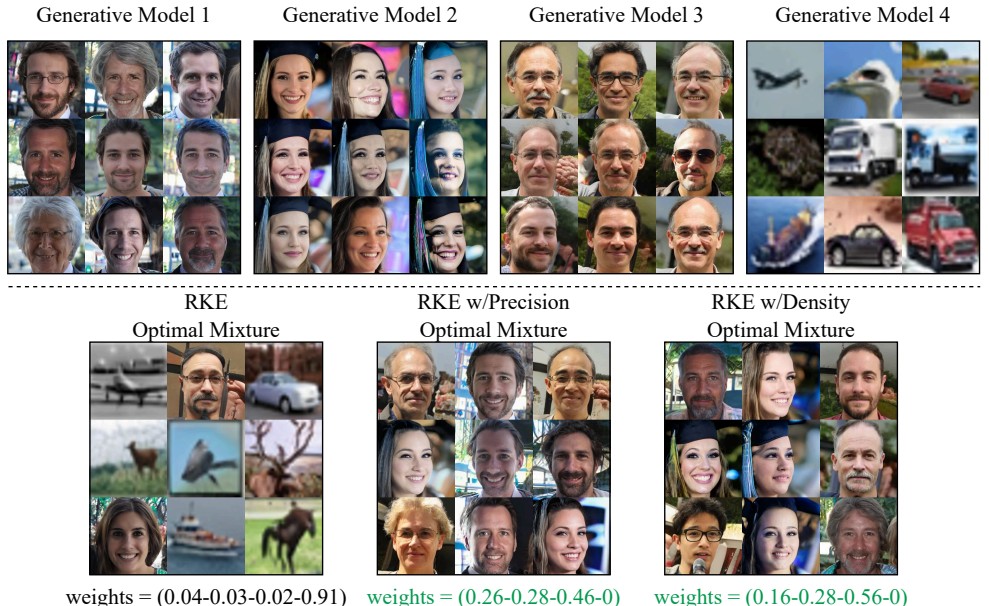

Figure 13: Visual demonstration of the effect of combining Precision/Density with RKE. The CI-FAR10 generator is excluded when these quality metrics are applied.

To address this, we incorporate a quality metric, specifically Precision/Density, into the optimization. We subtract the weighted Precision/Density from the RKE value, ensuring a balance between quality and diversity. The weight for the quality metric ($\lambda = 0.2$) was selected based on validation data to ensure comparable scaling between the two metrics. As a result, Figure 13 shows that the fourth arm, which had low-quality outputs, is assigned a weight of zero.

We use the RKE score as **K** and the weighted Precision/Density as **f** according to equation 5. The online algorithms were run for 4,000 steps, with the results depicted in Figure 5.

