# OpenReview forum: "Be More Diverse than the Most Diverse: Optimal Mixtures of Generative Models via Mixture-UCB Bandit Algorithms"
_ICLR.cc/2025/Conference — ICLR 2025 Poster_

### Official Review · Reviewer_nCW5 · 2024-11-04

**Soundness:** 3
**Presentation:** 3
**Contribution:** 3
**Rating:** 6
**Confidence:** 4

**Summary:**

This paper study online selection for generative models, in order to generate diverse samples. The authors formulated the problem as a mixture multi armed bandit problem and developed two algorithms for that: Mixture-UCB-CAB and Mixture-UCB-OGD. The authors developed theoretical guarantees for the Mixture-UCB-CAB algorithm. The authors conduct many experiments to show the efficacy of their developed methods.

**Strengths:**

It's interesting to see the authors formulated the generative model selection problem as an online selection problem. The authors also developed two algorithms for this new setting and provide theoretical guarantees for one of them. Experimental results demonstrate the efficacy of the proposed algorithms.

**Weaknesses:**

1. Since this is a new problem, can authors provide more motivations for online selection of generative models, e.g., how important is the ability to generate diverse samples? And how important is to save samples in the selection process.
2. The authors provide a convergence guarantee for Mixture-UCB-CAB in Thm 2. For comparison, what is the rate of convergence for the offline approach that randomly generate $T$ samples and then optimize over $\alpha$?
3. Does Thm 1 holds for all $\alpha$? Also, the guarantee in Thm 2 doesn't suffer the curse of dimensionality even if the algorithm is selection $\alpha \in R^m$; can authors explain why does that happen?
4. Compared to standard bandit problem where one gets an intermediate regret term at each round, it seems that the studied problems gets $O(t)$ (averaged) terms (the first Eq in Section 5), and all these terms are related to the previous selections $x_1, \cdots, x_{t-1}$. Can authors elaborate how do they deal with these terms in the analysis? What are some technical contributions?

**Questions:**

See above.

---

> ### Author Response · Authors · 2024-11-22
>
> We thank Reviewer nCW5 for the thoughtful feedback. We are pleased to hear that Reviewer nCW5 finds the formulation of the problem interesting. Please find our responses below.
>
> **1.1-Motivations behind online selection.**
>
> We argue that the combination of generative models is inherently an online problem. Please refer to "Motivations behind the online selection of generative models" in our general response on the top of the page for details.
>
> **1.2-Importance of the ability to generate diverse samples.**
>
> Diversity has been an important criterion in the evaluation of generative models in the literature. The well-known evaluation scores of Recall and Coverage have been proposed to exclusively assess the diversity of a generation model. The evaluated diversity scores can be used to ensure the model's sufficient grasp of the contents in the data distribution.
>
> **1.3-Importance of saving samples in the selection process.**
>
> Note that generating high-resolution image and video samples would be computationally and financially expensive. Our goal is to identify the optimal mixture of generation models by using the minimum number of queries to sub-optimal generative models. This can save the unnecessary costs of creating samples from weaker models to detect their lack of optimality.
>
>
> **2-Comparing Theorem 2 to offline approach.**
>
> Note that an offline approach will also have a worst-case error approximately $O(\sqrt{1/T})$. For conventional multi-armed bandit without the quadratic kernel term, Bubeck and Cesa-Bianchi (2012) showed a minimax lower bound on the regret per round that scales like $\sqrt{1/T}$ as $T$ increases, so Theorem 2 in our paper is tight within a logarithmic factor (this discussion is added to the updated paper after Theorem 2). This $\sqrt{1/T}$ order of growth of the error is also applicable to offline approaches. If we have two arms where the difference between their average rewards is $\sqrt{1/T}$, we cannot reliably tell which arm is better even if we are given the $T$ samples in an offline manner, and choosing the wrong arm will result in a $\sqrt{1/T}$ gap from the optimum. This shows the tightness of Theorem 2, in the sense that even if we do not have the quadratic kernel term, and even if $T$ samples are given to us in an offline manner, there is still a $\sqrt{1/T}$ error in the worst case, so Theorem 2 is tight within a logarithmic factor.
>
> **3-About Theorems 1 and 2, $\alpha$ and the curse of dimensionality.**
>
> Theorem 1 holds for any *fixed* $\alpha$. We also have a bound in Lemma 1 in the appendix which is a worst-case bound that holds simultaneously for every $\alpha$, which is a key step in the proof of Theorem 2. We will clarify this in the revised paper.
>
> The worst-case bound in Lemma 1 does not suffer from the curse of dimensionality, even though the bound holds simultaneously for every $\alpha$ and every possible sequence $n_1,\ldots,n_m$, where $n_i$ is the number of times arm $i$ is pulled up to the current round. This is because each quadratic term $\hat{\mathbf{K}}_{i,j}$ only depends on two models $i$ and $j$, so we only require taking union bound over choices of $n_i,n_j$ ($O(T^2)$ number of choices), rather than over choices of the whole sequence $n_1,\ldots,n_m$ ($O(T^m)$ number of choices). The quadratic structure of the loss function helps us avoid the curse of dimensionality.
>
> Theorem 2 does not suffer from the curse of dimensionality. The right hand side of Theorem 2 scales with $m$, but not exponentially. This is because we only require sufficient samples from the $m$ models in the worst case in order to obtain a good estimate. Therefore, we only need a sample size $T$ that is approximately proportional to $m$ in the worst case, as shown in Theorem 2. We do not require the samples to cover a large $m$-dimensional space.
>
> **4-About the analysis on the dependency between the selections.**
>
> The dependency between $x^{(1)},\ldots,x^{(T)}$ is indeed a major challenge in the analyses. The samples $x^{(s)},x^{(t)}$ are dependent, so $\kappa(x^{(s)},x^{(t)})$ does not have a tractable distribution. We bound the dependency between $x^{(s)},x^{(t)}$ via the chain rule of mutual information and Pinsker's inequality in the proof of Theorem 2. Intuitively, each $x^{(t)}$ may depend on $x^{(1)},\ldots,x^{(t-1)}$, but cannot significantly depend on every one of $x^{(1)},\ldots,x^{(t-1)}$ since the mutual information between $x^{(t)}$ and $x^{(1)},\ldots,x^{(t-1)}$ is bounded by the entropy of the choice of the arm, which is upper-bounded by $\log m$. This allows us argue that $\kappa(x^{(s)},x^{(t)})$ cannot significantly deviate from the situation where $x^{(s)},x^{(t)}$ are independent, and $T^{-2} \sum_{s,t \in [T]} \kappa (x^{(s)},x^{(t)})$ can be approximated by $T^{-2} \sum_{s,t \in [T]} \mathbf{K}_{i_s, i_t}$ where $i_t$ is the arm pulled at round $t$.

---

> > ### Comment · Reviewer_nCW5 · 2024-11-24
> >
> > I'd like to thank the authors for their responses. I'll keep my current rating.

---

> > > ### Author Response · Authors · 2024-11-29
> > >
> > > We thank Reviewer nCW5 for the feedback on our response.

---

### Official Review · Reviewer_mqps · 2024-11-07

**Soundness:** 3
**Presentation:** 4
**Contribution:** 2
**Rating:** 6
**Confidence:** 3

**Summary:**

The main goal of this work is to maximize the diversity of generated samples by selecting not a single but a mixture of generative models. Formulating first a population loss of quadratic form that can translate into evaluation scores including kernel inception distance (KID) and Renyi kernel entropy (RKE), this article proposes two online algorithms based on continuum-armed bandit and gradient descent, to find the optimal mixture through minimizing an upper confidence bound of the quadratic population loss.  Experiments show that the proposed algorithms are efficient at approaching the optimal mixture.

**Strengths:**

* This paper is well written and easy to follow.

* The theoretical framework underlying the proposed algorithms is well grounded.

* Extensive experiments were carried out to demonstrate the performance of the proposed algorithms.

**Weaknesses:**

* According to the literature review of this article, there seems to be little interest in finding a good mixture of different generative models. Indeed, if the goal is to approach the target distribution, it makes more sense to select the single best generative model than to use a mixture of different generative models, which are usually trained in an independent manner, therefore unlikely to complement each other.

* It is true that when the objective is to find the single best generative model, the online approach can help prevent sampling from suboptimal models. However, as using a mixture of generative models requires sampling from all member models, the online approach seems to be less useful in this setting.

**Questions:**

* Can the authors find some other works that also aim to find good mixtures of generative models, and compare their method to these works?

* Can the authors provide the quality scores Density (Naeem et al., 2020). and Precision (Kynkaanniemi et al., 2019) in the experiments that they conducted?

* Small question regarding Lines 257&259: is $\hat{L}(\mathbf{a};\mathbf{x}^{(t)})-(\mathbf{\epsilon}^{(t)})^{\rm T}\mathbf{a})$ a lower or upper bound of $L(\mathbf{a})$?



=======================================================================================================

**Update after rebuttal**

I would like to apologize for an important error that I made in the example presented in my last comment to the authors. I was so rushed to post this last comment before the discussion deadline to give the authors a chance to respond that I did not get to double check my math. As there is no other means for me to reach out to the authors, I have to rectify this error for the authors' information by editing my official review (at the suggestion of AC).

The error, which I realized after the deadline, came from the application of the FID formula. Under the assumption of Gaussian distributions, the FID is measured by
$$d_{\rm F}\left(P_r,P_g\right)=\Vert\mu_r-\mu_g\Vert^2+{\rm tr}\left(\Sigma_r+\Sigma_g-2(\Sigma_r\Sigma_g)^{\frac{1}{2}}\right)$$
where $\mu_r,\mu_g$ and $\Sigma_r,\Sigma_g$ are respectively means and covariances of $P_r,P_g$.

As in my example, the covariances of the real distribution $\mathcal{N}(\mu_r,I)$ and the two generative models $\mathcal{N}(\mu_g,I),\mathcal{N}(\mu_{\tilde{g}},I)$ are identity matrices, the FIDs are simply distances between the means:
$$d_{\rm F}\left(\mathcal{N}(\mu_r,I),\mathcal{N}(\mu_g,I)\right)=\Vert\mu_r-\mu_g\Vert^2=\Vert\epsilon_g\Vert^2,$$
$$d_{\rm F}\left(\mathcal{N}(\mu_r,I),\mathcal{N}(\mu_{\tilde{g}},I)\right)=\Vert\mu_r-\mu_{\tilde{g}}\Vert^2=\Vert\epsilon_{\tilde{g}}\Vert^2.$$

The error occurred when I calculated the FID between $\mathcal{N}(\mu_r,I)$ and a mixture $\mathcal{M}=\alpha\mathcal{N}(\mu_g,I)+\tilde{\alpha}\mathcal{N}(\mu_{\tilde{g}},I)$ of the two generative models. While the mean of $\mathcal{M}$ is indeed $\alpha\mu_g+\tilde{\alpha}\mu_{\tilde{g}}$ as I said in my comment to the authors, the covariance, however, should be $I+\alpha\tilde{\alpha}(\mu_g-\mu_{\tilde{g}})(\mu_g-\mu_{\tilde{g}})^{\rm T}$, **not** $I$.

Therefore, $d_{\rm F}\left(\mathcal{N}(\mu_r,I),\mathcal{M}\right)\approx\Vert\mu_r-\alpha\mu_g-\tilde{\alpha}\mu_{\tilde{g}}\Vert^2+\alpha\tilde{\alpha}\Vert\mu_g-\mu_{\tilde{g}}\Vert^2$ for large $\Vert\mu_g-\mu_{\tilde{g}}\Vert^2$,
leading to
$$d_{\rm F}\left(\mathcal{N}(\mu_r,I),\mathcal{M}\right)\approx\Vert\alpha\epsilon_g+\tilde{\alpha}\epsilon_{\tilde{g}}\Vert^2+\alpha\tilde{\alpha}\Vert\epsilon_g-\epsilon_{\tilde{g}}\Vert^2.$$Then we have, for independent $\epsilon_g,\epsilon_{\tilde{g}}$, that
$$E\\{d_{\rm F}\left(\mathcal{N}(\mu_r,I),\mathcal{M}\right)\\}\approx\alpha^2E\\{\Vert\epsilon_g\Vert^2\\}+\tilde{\alpha}^2E\\{\Vert\epsilon_{\tilde{g}}\Vert^2\\}+\alpha\tilde{\alpha}E\\{\Vert\epsilon_g\Vert^2\\}+\alpha\tilde{\alpha}E\\{\Vert\epsilon_{\tilde{g}}\Vert^2\\}=\alpha E\\{\Vert\epsilon_g\Vert^2\\}+\tilde{\alpha}E\\{\Vert\epsilon_{\tilde{g}}\Vert^2\\},$$
which is always minimized by taking the single model with the smallest error.

The intuitive reasoning behind this example is that a mixture of Gaussians is unlikely to match better the target Gaussian distribution than (the best) single Gaussians. The lack of discussion on the limitations and the risks of the mixture approach is my main criticism of this work.

---

> ### Author Response · Authors · 2024-11-22
>
> We thank Reviewer mqps for the thoughtful feedback. We are pleased to hear that Reviewer mqps finds the paper well written and easy to follow. Please find our responses below.
>
> **1-Motivation of finding a good mixture of different generative models.**
>
> While it is true that prior works focus on finding a single best model, we hope that our methods can open a new avenue of combining generative models. Our experiments show that our methods can select a mixture of models that outperforms any single one of those models in terms of diversity. Also, we believe that the models being trained in an independent manner is beneficial to our methods, since a mixture of independent samples will likely improve diversity. This is indeed the case for large-scale text-to-image models, which are usually trained on different training datasets. Figure 1 in the revised paper shows one example where three standard text-to-image models generate differently-styled cartoon giraffe pictures. In such cases, considering the mixture of generative models can significantly add to the diversity of output data.
>
>
>
> **2-Usefulness of the online learning approach.**
>
> Note that the optimal mixture will not necessarily involve every model. An online selection algorithm will prevent the models that are not in the optimal mixture (or has a low percentage in the optimal mixture) from being sampled frequently. Also, the problem of combining generative models is online in nature. Please refer to "Motivations behind the online selection of generative models" in our general response on the top of the page for details.
>
> Also, we have a sparse version of our algorithm presented in Appendix 8.2, which attempts to choose a mixture involving a small subset of the models. In this setting, we have to avoid sampling too frequently from suboptimal models not present in the optimal mixture. An online algorithm can discard a model as soon as we can confidently tell that it is suboptimal.
>
>
> **3-Other works on mixtures of generative models.**
>
> To the best of our knowledge, our work is the first to propose the *selection of a mixture of the distributions* of multiple generative models. Specifically, we aim to highlight the possible improvements in the diversity of generated data by using a mixture of several generative models.
>
> **4-About Density and Precision.**
>
> We have evaluated the Precision, Density (quality scores) as well as recall, coverage (diversity scores) for the KID-based experiments on the real image datasets LSUN-bedroom and FFHQ. We note that these scores cannot be used in our experiments on images generated by text-to-image models, because the reference dataset needed by the scores is unknown.
>
> We have included Table 1 and the following explanations in the appendix of the updated paper. In our quantitative evaluation, we observed that the Precision of the optimal mixture is similar to that of the maximum Precision score among individual models. On the other hand, the Recall-based diversity improved in the mixture case. However, the quality-measuring Density score slightly decreased for the selected mixture model, as Density is a linear score for quality that could be optimized by an individual model. On the other hand, the Coverage score of the mixture model was higher than each individual model.
>
> Note that Precision and Density are scores on the average quality of samples. Intuitively, the quality score of a mixture of models is the average of the quality score of the individual models, and hence the quality score of a mixture cannot be better than the best individual model. On the other hand, Recall and Coverage measure the diversity of the samples, which can increase by considering a mixture of the models. To evaluate the net diversity-quality effect, we measured the FID score of the selected mixture and the best individual model, and the selected mixture model had a better FID score compared to the individual model with the best FID.
>
> **5-"Is $\hat{L}(\mathbf{a};\mathbf{x}^{(t)})-(\mathbf{\epsilon}^{(t)})^{\rm T}\mathbf{a})$ a lower or upper bound of $L(\mathbf{a})$?"**
>
> It is a lower bound. In the conventional multi-armed bandit setting, the goal is to maximize the reward, so an upper confidence bound is used. In comparison, our goal is to minimize the loss function $L(\mathbf{a})$, so we require a lower confidence bound. We still use the term "upper confidence bound" to conform with standard terminology. (Although we can flip the sign and consider the negative loss function to be the reward and use an upper confidence bound, this will make the reward always negative and is a negative definite function, which is unnatural.)

---

> > ### Comment · Reviewer_mqps · 2024-11-28
> >
> > I thank the reviewers for their clear reply, which adequately addressed several of my questions. As reflected in my initial review, I appreciate the presentation quality and the extensive experimentation of this paper. My biggest concern is about the interest of improving diversity by selecting a mixture of generative models, which is not entirely resolved by the authors' reply.
> >
> > First what I meant by "trained in an independent manner" is not "trained on independent samples", but *trained by independent optimisation formulations as opposed to a joint optimisation*.  My reasoning is that if the target distribution is a Gaussian mixture model (GMM), combining single models that generate from **different** Gaussian components in the target GMM would make perfect sense, however it is unlikely that single generative models happen to match different components unless they are trained through a joint optimization with a penalty that encourage them to learn different Gaussian components.
> >
> > This is why I raised the question about the quality score: using a mixture of generative models that normally do not complement each other probably does not give a better approximation of the target distribution. As confirmed by the authors, using a mixture of generative models has little benefit in improving the quality. In contrary, it can lead to a degraded quality compared to the best single model.
> >
> > Then I think the value of this contribution depends crucially on whether it has practical interest to favor diversity over quality when evaluating generative models. Unless the authors could provide stronger arguments on this point, my current stand on this paper is towards rejection.

---

> ### Author Response · Authors · 2024-11-29
>
> We thank Reviewer mqps for the thoughtful feedback on our response. We are glad to hear that our previous response addressed several of the reviewer's questions. Regarding the reviewer's remaining comments, we would like to raise the following:
>
> - We respectfully disagree with the assertion that "using a mixture of generative models that normally do not complement each other probably does not give a better approximation of the target distribution". For example, Figure 2 in the updated paper (was Figure 1 in the old version) shows that for models trained on the FFHQ and LSUN-Bedroom datasets, our mixture model (Mixture-UCB-CAB and OGD) offers a better approximation than each individual model (One-Arm Oracle is the best individual model), where the goodness of approximation is measured by KID. Tables 1 and 2 in the updated paper also suggest that our mixture model can offer an improvement in terms of the FID score. Also, note that quality scores such as Density and Precision are not measures of goodness of distribution approximation themselves, as they are merely measuring the average quality of samples (one cannot conclude that two distributions are similar merely because their expected values are similar). A measure of goodness of distribution approximation (e.g., FID and KID) has to take diversity into account to capture any mismatch between the higher-order moments (e.g. the covariance matrix) of the distributions.
>
> - We are not advocating that we should favor diversity over quality, as we believe both of them are important. Standard scores such as FID and KID naturally take both quality and diversity into account. In the numerical results in Tables 1, 2 and Figure 2 in the updated paper, we observe that the FID and KID scores of the proposed mixture model improve upon the scores of each individual generative model. This shows that the proposed mixture model not only improves diversity, but is also generally favorable in terms of minimizing FID and KID.
>
> - The reason why this paper emphasizes on diversity is because this is the main improvement provided by the proposed mixture model. As demonstrated by our experimental results (Tables 1, 2 and Figure 2 in the updated paper), the improvement in diversity is so significant that the standard scores (FID and KID, which take both quality and diversity into account) also improve. Our method is suitable not only to users who favor diversity, but also to users who value both diversity and quality. (Admittedly, the mixture model does not offer improvement to users who only value quality, but this position is uncommon, since only valuing quality would mean that the users prefer a mode-collapsed model that always produces the same high-quality sample.)
>
> - Our Mixture-UCB algorithms are designed with both quality and diversity in mind. The loss function in equation (4) includes both a linear quality term $\mathbb{E}\bigl[f(X)\bigr]$ and a quadratic diversity term $\mathbb{E}\bigl[\kappa(X,X')\bigr]$ (the linear term and the quadratic term indeed correspond to quality and diversity for KID in equation (3)). Our method is flexible, and the relative weights of quality and diversity can be adjusted by the user. In the extreme scenario where the user assigns zero weight to the diversity term (i.e, the user only considers the quality factor), the Mixture-UCB algorithm will reduce to applying standard UCB to optimize the quality score and, as guaranteed by Theorem 2, will converge to the generative model with the maximum quality score. However, this scenario would be uncommon in practice as a typical user will likely consider both diversity and quality factors.

---

> > ### Comment · Reviewer_mqps · 2024-12-01
> >
> > I thank the authors for their reply, and would like to raise some follow-up questions.
> >
> > - In Figure 2 of the revision, we can see that using mixture of models is particularly effective on FFHQ truncated generators that "generate diversity-controlled images centered on eight randomly selected points". I think this supports my point that "generative models normally do not complement each other" unless they were trained in a joint manner such as the FFHQ truncated generators in Figure 2. To me, this is an important point to be discussed in the paper as a potential limitation of the proposed mixture approach. Also could the authors explain why they think Density and Precision only compare the expected values between the generative model and the target distribution? And if the authors believe that Density and Precision are "not measures of goodness of distribution approximation themselves", why not use other metrics to access the quality of generated data?
> >
> > - I do find the improvement in FID and KID very interesting, which is the main reason of my positive rating in the initial review.
> >
> > - My concern is not exactly about the focus on the improvement of diversity, but about the improvement of diversity (potentially) at the cost of quality. This is why I particularly asked about the quality scores.
> >
> > - I agree that the linear quality term $\mathbb{E}\bigl[f(X)\bigr]$ allows a control on the quality, however it only compares the expected values, which, as pointed out by the authors, can be insufficient. Moreover, this optimization formulation does not automatically guarantee that mixtures of generative models give better values on the linear quality term $\mathbb{E}\bigl[f(X)\bigr]$ than single models, and the paper does not seem to provide empirical evidence on this point.

---

> ### Author Response · Authors · 2024-12-02
>
> We thank Reviewer mqps for the feedback on our response. Please find our responses below.
>
> - We respectfully disagree that "generative models normally do not complement each other" is a potential limitation of the proposed mixture approach. Figures 2(a) and 2(b) show that the mixture approach outperforms each of the individual popular generative models applied on standard datasets, without truncation. For Figures 2(a) and 2(b), we did not apply any special treatment to make the models complement each other. As we stated in the text, the mentioned generative models are pre-trained and made available by the well-cited dgm-eval repository (Stein et al, NeurIPS 2023), which are not trained jointly to fit our mixture approach.
>
> - Using FID and KID for evaluating generative models is a standard approach. We believe that our method outperforming individual models in terms of FID and KID (while having acceptable quality scores such as Precision and Density) is sufficient to show the merits of our method.
>
> - Regarding the concern that our approach results in sub-optimal quality scores such as Precision and Density, we emphasize that the Precision and Density scores of our approach are not poor. In some situations, we can even improve upon FID without reducing Precision. For example, in Table 2, our approach ranks 1st (tied) in Precision and 2nd in Density among 5 models (while being 1st in FID and Coverage, and 2nd in Recall), giving an almost overall improvement upon the best individual model. In Table 1, our approach ranks 4th among the 6 models for Precision and Density (while being 1st in FID and Recall).
>
> - While the reviewer's concern appears to be based on the Precision and Density scores, we remark that these scores are not measures of goodness of distribution approximation, and are **maximized by a mode-collapsed generative model** that always outputs the same high-quality sample. A reasonable model almost always has lower Precision and Density scores compared to the mode-collapsed model. Therefore, not maximizing the Precision and Density scores is not an indicator of weakness. As also emphasized by (Sajjadi et al, NeurIPS 2018) and (Naeem et al, ICML 2020) who proposed the scores, Precision and Density should be examined together with Recall and Coverage to provide a holistic evaluation of a generative model.
>
> To answer the specific questions of the reviewer:
>
> - **Density, precision and expected value.** To see why Density and (improved) Precision (Kynkäänniemi et al, NeurIPS 2019) are expected values, note that their definitions involve an average over the generated samples, in the form $\frac{1}{M}\sum_{i=1}^M f(x_i)$ where $x_1,\ldots,x_M$ are the generated samples and $f$ is a certain function. Therefore, they are in the form $\mathbb{E}[f(X)]$ where $X$ is a random generated sample. An average can always be maximized by a degenerate distribution (a mode-collapsed model) at the point $x$ that maximizes $f(x)$.
>
> - **Regarding "why not use other metrics to access the quality of generated data".** We use FID and KID to assess our method. Unlike Density and Precision, FID and KID are mathematical (pseudo)metrics to measure the distance between probability distributions, and quantify how close the distribution of the generated samples is to the true distribution. In the case of KID (with a universal kernel function, e.g. Gaussian kernel), the distance is zero if and only if the reference and generative model's distributions are the same.
>
> - To address a potential confusion, note that there are two notions that are termed "quality": the average quality of samples as measured by Precision and Density, and the quality of the whole generative model (i.e., how well the model approximates the true distribution) as measured by FID and KID. The average quality of samples is only one aspect of the quality of the whole model. Here, we are using the word "quality" to mean the average quality of samples (as in Sajjadi et al, 2018), and use "goodness of distribution approximation" to mean the quality of the whole model.

---

> ### Comment · Reviewer_mqps · 2024-12-03
>
> I thank the authors for their reply. Below are some follow-up remarks.
>
> **Measures that only compare the expected values of two distributions**
>
> I thought what the authors meant by saying that Precision and Density only compare "expected values" of the real distribution $P_r$ and the generative model $P_g$ is that these two measures only take into account the means $E_{x_r\sim P_r}[x_r]$ and $E_{x_g\sim P_g}[x_g]$. According to the reference (Kynkäänniemi et al, NeurIPS 2019) pointed out by the authors, it does not seem to be the case for Precision defined in (1) of this paper.
>
> **Complementarity of generative models**
>
> As we keep coming back to my statement "generative models normally do not complement each other", I think it is better that I try to explain with a concrete example. In this example, we will use FID and consider that the real distribution is a normal distribution $\mathcal{N}(\mu_r, I)$ of unknown mean $\mu_r$ and known identity covariance  (the knowledge of identity covariance is assumed to simplify the discussion). Let $\mathcal{N}(\mu_g, I), \mathcal{N}(\mu_{\tilde{g}}, I)$ be two generative models obtained in an independent manner so that their errors $\epsilon=\mu_r-\mu_g, \tilde{\epsilon}=\mu_r-\mu_{\tilde{g}}$ are independent as well. It is easy to see that the expected value of a mixture $\mathcal{M}=a\mathcal{N}(\mu_g, I)+\tilde{a}\mathcal{N}\mu_{\tilde{g}}, I)$ is given by $a\mu_g+\tilde{a}\mu_{\tilde{g}}$, which is equal to $\mu_r+a\epsilon+\tilde{a}\tilde{\epsilon}$. Then we have ${\rm FID}=\Vert a\epsilon+\tilde{a}\tilde{\epsilon}\Vert^2$. As $E[{\rm FID}]=a^2E[\Vert\epsilon\Vert^2]+\tilde{a}^2E[\Vert\tilde{\epsilon}\Vert^2]$, ${\rm FID}$ tends to be minimized by taking the best single model with the smallest error, specially in very high dimensions where ${\rm FID}\simeq E[{\rm FID}]$ as a consequence of $\epsilon^{\rm T}\tilde{\epsilon}\simeq 0$.
>
> Of course, many real data can not be well approximated by a single normal distribution, specially when they are multimodal. This is why I mentioned the case where the target distribution is a Gaussian mixture. Let us consider now that the real distribution is a mixture of two Gaussian components $\mathcal{N}_1,\mathcal{N}_2$. If we have two generative models $g,\tilde{g}$ with $g$ performing better w.r.t. $\mathcal{N}_1$ and $\tilde{g}$ w.r.t. $\mathcal{N}_2$, then combining them would probably lead to a better approximation of the target Gaussian mixture. However, when $g$ happens to perform better for both $\mathcal{N}_1$ and $\mathcal{N}_2$, using a mixture of $g,\tilde{g}$ is likely to result in a decreased quality of generated samples. I think both cases are probable scenarios in practice, which is what I referred to as a "potential limitation" of the mixture approach.

---

> > ### Author Response · Authors · 2024-12-03
> >
> > We thank Reviewer mqps for the feedback on our response. Please find our responses below.
> >
> > **1- About Density and Precision**
> >
> > We have clarified in our last response that what we meant was that Density and Precision are in the form $\mathbb{E}[f(X)]$, which is an expected value of a function $f$. We cannot conclude that $X_g$ has a similar distribution as $X_r$ only by looking at $\mathbb{E}[f(X_g)]$ and $\mathbb{E}[f(X_r)]$. FID and KID are more suitable for measuring distances between probability distributions.
> >
> > For example, for Precision (eqn. (1) in Kynkäänniemi et al. 2019), the function $f$ is taken to be $f(x) = 1$ if $x$ is in the manifold of the real images (defined via k-th nearest neighbor by Kynkäänniemi et al), and $f(x) = 0$ otherwise. We can see that Precision $\mathbb{E}[f(X_g)]$ can attain its maximum value $1$ when the model is mode-collapsed and always output the same sample that lies in the manifold of the real images. Therefore, a high Precision does not indicate that the model approximates the true distribution well.
> >
> >
> > **2- About normal distribution and FID**
> >
> > The FID formula provided by the reviewer actually attains its optimal value at a mixture distribution. According to the formula by Reviewer mqps, $$\mathbb{E}[{\rm FID}]=a^2\mathbb{E}\bigl[\Vert\epsilon\Vert^2\bigr]+\tilde{a}^2\mathbb{E}\bigl[\Vert\tilde{\epsilon}\Vert^2\bigl],$$
> > where $\tilde{a} = 1-a$. This quadratic function of $a$ is minimized at $$a = \frac{\mathbb{E}[\Vert\tilde{\epsilon}\Vert^2]}{\mathbb{E}[\Vert\tilde{\epsilon}\Vert^2] + \mathbb{E}[\Vert\epsilon\Vert^2]}$$
> > This represents a mixture distribution (unless one of $\mathbb{E}[\Vert\epsilon\Vert^2]$ and $\mathbb{E}[\Vert\tilde{\epsilon}\Vert^2]$ is zero, which is extremely unlikely since it means that one model is completely accurate).
> >
> > (To address a potential source of confusion, note that since $a$ and $\tilde{a}$ are mixture weights, we have $a +  \tilde{a} = 1$ instead of $a^2 + \tilde{a}^2 =1$. If we incorrectly assumed $a^2 + \tilde{a}^2 =1$, we would arrive at the incorrect conclusion that FID is minimized at a single model.)
> >
> > Therefore, even the reviewer's example shows that the FID-minimizing model will be a mixture of the models with non-zero weights. We hope that this example, together with the experiment results in Figure 2 and Tables 1,2, will finally convince the reviewer that "generative models normally do not complement each other" is not a weakness of the mixture approach.
> >
> > **Additional Note:** We believe the actual formula of FID (including the covariance term) is more complicated than the reviewer's provided formula. In this response, we adopt the reviewer's formula for the sake of simplicity, but would like to note that the similar conclusion that FID is generally minimized by a mixture model remains valid for the correct FID formula with the covariance term. This conclusion is also supported by our numerical results in Tables 1 and 2.

---

### Official Review · Reviewer_GUyR · 2024-11-12

**Soundness:** 3
**Presentation:** 3
**Contribution:** 3
**Rating:** 5
**Confidence:** 3

**Summary:**

In this paper, the authors focus on the online selection of generative models, and in particular, the optimal linear mixture among a set of such models.
The problem appears novel, and the authors make interesting connections to the maximization of some kernel-based scores and multi-armed bandit.
Based on this, the authors propose Algorithms 1 and 2 to solve this online selection of mixture efficiently, with performance guarantee given in Theorem 1 and 2, respectively. (Although I have some concerns on the settings and theoretical results, see below).
These methods can be used for widely used kernel inception distance (KID) and Renyi kernel entropy (RKE), and are tested on realistic image and text data in Section 6.

**Strengths:**

* The paper consider online selection of generative mixture models, which, to the best of knowledge, is a novel problem of interest.
* By making interesting connection to kernel-based scores and multi-armed bandit, the authors propose efficient methods to solve the above problem, with some theoretical guarantee.
* Experiments on realistic data are provided, showing the practical applicability of the proposed approaches.

**Weaknesses:**

* It would be great to discuss the limitation of the proposed approach, see below for my detailed comments/questions.
* some settings and theoretical results need clarification, see below for my detailed comments/questions.

**Questions:**

Below are a few questions and/or comments.

1. The problem appears novel, so I believe it makes sense to better motivative it. For example, in which context are we interested in picking a model to generate a sample at each round, why it is of interest to use "the fewest possible sample queries"? How the proposed method performs in an offline setting, with respect to performance and/or scalability?
2. When summarizing the contribution of this paper, could the authors also provide (forward) pointers to the precise results? For example, "proposing an online learning framework in Section ??". I personally believe that this may facilitate the interested readers to quickly grasp the main contribution of the paper.
3. Is the working assumption of linearly mixed model somewhat restrictive? Is there something else in the literature, or even such linear combination is (the first time) proposed by the authors in this paper? In fact, on the top row of Figure 3, there is a linearly mixtured "dog" that appears a bit bizarre: is this due to some limitation of this linear mixture?
4. I personally find Theorem 1 a bit surprising: To me, kernel matrix "estimation" problem plus some online selection problem, and solving the former problem in general requires a lot of samples to have a tight spectral norm control on the estimated kernel matrix. I believe that the authors avoid this issue by assuming/focusing on the case of bounded. Could the authors comment more on this? For example, does this bounded kernel/loss function setting limit the practical interest of the proposed methods? Also, could the authors comment on the observed sample size $n_i$ for the proposed OGD method to make sense? We do not see this in Theorem 2 and this has an impact on the computational complexity I believe?
5. a tiny side remark: Figure 3 appears in the main text but commented in the appendix.

---

> ### Author Response · Authors · 2024-11-22
>
> We thank Reviewer GUyR for the thoughtful feedback. We are pleased to hear that Reviewer GUyR finds the connection to kernel-based scores and multi-armed bandit interesting. Please find our responses below.
>
> **1-Context in which we are interested in picking a model to generate a sample at each round.**
>
> An online selection method where we pick a model at each round is a natural approach to combine generative models, that can reduce the cost of generating from sub-optimal models. Please refer to "Motivations behind the online selection of generative models" in our general response on the top of the page for details.
>
>
> **2-Forward pointers to the precise results.**
>
> Thank you for the suggestion. Pointers have been added to the updated paper.
>
> **3-"Is the working assumption of linearly mixed model somewhat restrictive?"**
>
> Note that we treat each generative model as a black box. We are not allowed to examine the architectures and parameters of the models in order to combine them. We are also not allowed to combine the pixels or embedding vectors of the images generated by different models (if we decide to pull an arm, we must use the image generated by the arm as is). Therefore, we can only combine them through choosing a mixture (i.e., assigning a percentage to each model, e.g., 70 percent of the samples come from Model 1, and 30 percent come from Model 2). We make this assumption for the sake of full generality. By treating each model as a black box, our methods can combine any set of models, including models with vastly different architectures, without any prior knowledge on how we can combine the parameters of different models in a reasonable manner. The use of linear mixture is a consequence of this general assumption on the models.
>
>
>
> **4-About Theorem 1.**
>
> We would like to clarify that the matrix $\mathbf{K}$ in the paper is not the usual $n \times n$ kernel matrix (where $n$ is the sample size), but rather the $m \times m$ "average kernel matrix" ($m$ is the number of models), where the entry $\mathbf{K}_{i,j}$ is the average of $\kappa(X,X')$ where $X$ is a random sample from model $i$ and $X'$ is a random sample from model $j$. Since $m$ is usually much smaller than $n$, the matrix $\mathbf{K}$ is significantly easier to estimate than the usual kernel matrix.
>
> Indeed, the bounds in Theorem 1 and 2 are possible because the loss function (4) is assumed to be bounded (see the beginning of Section 5.1). Please note that boundedness is not a big obstacle, since several popular kernels (e.g., Gaussian kernel) are bounded by default. In case the kernel function is unbounded (when the input data is unbounded), we can still compute a bound on the data in order to give a bound on the loss function.
>
> About the sample size needed by OGD, although Theorem 2 only applies to Mixture-UCB-CAB, we expect a similar $O(\sqrt{\frac{\log T}{T}})$ error to apply to OGD, since this is a general phenomenon for regret bounds that do not depend on the sample distributions of the arms (see updated discussions after Theorem 2). Nevertheless, proving a regret bound for OGD would be more challenging than CAB due to the difficulty of keeping track of the proportion vector $n^{(t)}/t$, so we leave the analyses of the sample size needed by OGD to future studies.
>
> **5-About Figure 3.**
>
> Thank you for pointing this out. This has been fixed in the updated paper.

---

### Official Review · Reviewer_pb52 · 2024-11-12

**Soundness:** 2
**Presentation:** 3
**Contribution:** 3
**Rating:** 6
**Confidence:** 3

**Summary:**

This paper aims to solve the online selection task over a group of well-trained generation models. It explores the selection of a mixture of multiple generative models and formulate a quadratic optimization problem to optimize the kernel-based evaluation scores including kernel inception distance (KID) and Renyi kernel entropy (RKE). Specifically, it proposes an online learning approach called Mixture Upper Confidence Bound (Mixture-UCB). Theoretically, regret analysis is provided for one method (Mixture-UCB-CAB). Experimental results illustrate the effectiveness of the proposed method for text-based and image-based generative models.

**Strengths:**

1. Overall, this paper is well-written and easy to follow.
2. The proposed method (Mixture-UCB) is somehow novel, although it is inspired by classical UCB in the multi-armed bandit setting.
3. Theoretical results about the regret bound are provided for the proposed Mixture-UCB-CAB. The proof seems right although I have not checked the proof line-by-line.
4. Empirical results illustrate the effectiveness of the proposed method in finding the optimal mixture of text-based and image-based generative models.

**Weaknesses:**

1. I am afraid that the online selection of well-trained generative models might have few applications because it is already costly for the (large) generative model inference, then why do we need online selection rather than batch selection? Discussions about practical applications can be added.
2. Experimental results show that Mixture-UCB-OGD might be better than Mixture-UCB-CAB. However, theoretical guarantees about Mixture-UCB-OGD are missing. I know it might be more challenging and more detailed discussions can be added to clarify why.

**Questions:**

In practice, FID metric is widely-used in the evaluation of generative models. Can this paper cover this metric and why?

---

> ### Author Response · Authors · 2024-11-22
>
> We thank Reviewer pb52 for the thoughtful feedback. We are pleased to hear that Reviewer pb52 finds our paper well-written and easy to follow. Please find our responses below.
>
> **1-Online selection of well-trained generative models might have few applications.**
>
> Large generative model inference being costly is precisely the motivation of our online approach. If we have several large models where generating one sample can take about 9 seconds (for the SD-XL model on one A100 GPU), then generating a batch of 1000 samples from each model to perform conventional score-based evaluation would take several hours. Instead, we should generate the samples one by one in an online manner, quickly ruling out the obviously suboptimal models, while generating more samples from the apparently better models. Please refer to "Motivations behind the online selection of generative models" in our general response.
>
>
>
> **2-Theoretical guarantees about Mixture-UCB-OGD.**
>
> The analysis of Mixture-UCB-OGD would be more challenging than that of Mixture-UCB-CAB, due to the difficulty of keeping track of the proportion vector $n^{(t)}/t$. The analysis of Mixture-UCB-OGD would involve a complicated dynamical system with the state being the proportion vector and the matrix $\hat{\mathbf{K}}$.
>
> **3-About FID metric.**
>
> We note that the FID metric can be decomposed into the sum of two terms: 1) a quadratic cost on the embedded means' difference norm $\Vert \mu_{G} -\mu_X \Vert_2^2$ and 2) a non-quadratic cost on the embedded covariances $\Vert \Sigma_G^{1/2} - \Sigma_X^{1/2}\Vert^2_F$. Finding a quadratic approximation of the second FID component will be an interesting future direction to extend our online evaluation method to the FID score.

---

> > ### Comment · Reviewer_pb52 · 2024-11-29
> >
> > Thanks for the response. The authors have addressed my concerns and I will keep the score.

---

> > > ### Author Response · Authors · 2024-11-29
> > >
> > > We thank Reviewer pb52 for the feedback on our response. We are pleased to hear that the response addressed the reviewer's comments.

---

### Official Review · Reviewer_x3ew · 2024-11-25

**Soundness:** 3
**Presentation:** 3
**Contribution:** 2
**Rating:** 6
**Confidence:** 3

**Summary:**

Given a group of generative models, this paper studies the problem of improving the diversity (and quality) of generated outputs by combining them into an (optimal) mixture. The authors present the Mixture-UCB framework, encompassing two specific algorithms, Mixture-UCB-CAB and Mixture-UCB-OGD, designed by iteratively optimizing a quadratic objective (wrt the mixture weights) on kernel-based eval metrics and efficiently formulating the mixture of models under an online bandit setting. Specific metrics include Kernel Inception Distance (KID) and Rényi Kernel Entropy (RKE). Theoretical regret bounds have provided adequate support for Mixture-UCB-CAB, and experimental evaluations demonstrate the advantages of both algorithms across various datasets and model types.

**Strengths:**

1. The paper is generally well-written and well-structured, with clear definitions and visualizations.
2. The focus on mixtures of generative models to achieve superior diversity (and quality) appears innovative and addresses a limitation in traditional model selection approaches, which aim to find only a single best-performing model. Being able to customize the support size of the mixture is a good plus.
3. The theoretical analysis for Mixture-UCB-CAB is well-formulated and provides near-optimal guarantees (i.e., up to logarithmic factors of $m$ and $T$).
4. The diverse experiments demonstrate the algorithms' practical applications and performance gains, especially in exciting domains such as text-to-image generation.

**Weaknesses:**

1. While Mixture-UCB-OGD seems computationally more efficient than Mixture-UCB-CAB, the absence of a theoretical guarantee akin to Theorem 2 for CAB leaves an open question about its convergence and reliability.
2. Linear mixtures show their ability to enhance diversity. Still, the data distributions might produce mixtures that lack coherence, as some of the visual examples hint at (e.g., in Figure 3, the mixture model generated both realistic and unrealistic car images). In other words, optimizing the single diversity metric may not capture users' needs (e.g., a model that can generate images of cars with different sizes, poses, coloring styles, and backgrounds may be more natural to be said "diverse").
3. The mixture model approach may not be suitable for memory-efficient use cases, such as deployment on end devices like smartphones or smart home modules. Storing, updating, and switching among multiple generative models (e.g., this might require loading new parameters into memory) could significantly increase memory requirements and other costs, making the approach impractical for some critical applications. A possible mitigation strategy could be distilling a mixture of large models into a single, smaller-scale model, thereby retaining the benefits of the mixture while reducing resource needs.

**Questions:**

Could the authors kindly consider the weaknesses highlighted above and share any thoughts, feedback, or responses they might have? Also, I wonder about the typical scenarios where the mixture models fail to improve diversity or quality.

---

> ### Author Response · Authors · 2024-11-27
>
> We thank Reviewer x3ew for his/her thoughtful feedback. We are glad to hear that Reviewer x3ew finds our paper "well-written" and "well-structured". Please find our responses below.
>
> **1-Theoretical guarantee for OGD.**
>
> Proving a theoretical guarantee for Mixture-UCB-OGD would be more challenging compared to Mixture-UCB-CAB. A theoretical guarantee for Mixture-UCB-OGD would require analyzing a complicated dynamical system, where the state is the proportion vector and the matrix $\hat{\mathbf{K}}$. Therefore, we only give a theoretical guarantee for Mixture-UCB-CAB in this paper, and instead rely on experimental results to show the performance of Mixture-UCB-OGD. Theoretical guarantees for Mixture-UCB-OGD are left for future studies.
>
>
> **2-Coherence of generated samples.**
>
> While Reviewer x3ew has raised an interesting point of distinguishing diversity of styles (e.g., realistic and unrealistic, where some may regard as a lack of coherence in style) and diversity of the depicted objects (e.g., cars of different sizes), the choice of which kind of diversity to prefer seems to be rather subjective. A user who prefers diversity in terms of coloring styles (as mentioned by the reviewer) may also prefer diversity in terms of realistic/unrealistic styles.
>
> In our method, the quantification of diversity using RKE and KID scores depends on the choice of embedding model used in the evaluation process. In our experiments, we adopted the standard DINOv2 model to embed image samples, following the recommendation of Stein et al. in [1]. The results indicate that DINOv2 embeddings recognize cartoonish styles as a form of diversity. We note that the proposed method is adaptable and can be applied with alternative embedding models or custom features that represent the aspects of diversity most relevant to the user’s goals. Using a different embedding model would shift the emphasis in diversity quantification to the styles and features prioritized by the new embedding, potentially aligning more closely with the user's preferences.
>
>
> **3-About memory efficiency.**
>
> To run our algorithm locally with no access to external computing resources, we indeed require loading all the generative models. However, the strength of our algorithm is more apparent in a remote setting, where the user sends requests to several online services that host the generative models. Our algorithm only requires black-box access to the generative models, and hence it can be applied on all the popular commercial generative models. In the remote setting, the memory usage is much smaller since the user only needs to store the generated samples, not the whole models.
>
> The reviewer's suggestion of distilling a mixture of large models into a single small model is very interesting. Nevertheless, we note that it is rather different from the black-box approach in our paper. Distilling large models using black-box access likely requires a significantly large number of samples. If we require white-box access to the models (e.g., the weights of the neural networks), this can limit the application of the method, since it may be difficult to combine models with significantly different architectures in this manner, and white-box access is unavailable for proprietary models. Also, a small distilled model may require more computational resources than the aforementioned remote setting.
>
> In sum, the black-box approach of our method has two major advantages: applicability to general (open-source and proprietary) generative models with different architectures, and applicability to the remote setting suitable for devices with limited computational resources. We have added some discussions related to this point to the updated introduction.
>
>
> **4-Typical scenarios where the mixture models fail to improve diversity or quality.**
>
> The effectiveness of using a mixture of generative models depends on their probability distributions. When all the models represent the same distribution, the potential benefit of combining them is limited. However, if the models generate samples from different distributions, their mixture can lead to an improvement in diversity scores. This is particularly relevant for state-of-the-art prompt-based generative models, where variations in architecture and training data often result in different probability distributions across the trained models. As shown in Figure 1 of the revised introduction, three standard text-to-image models produce visually different samples for the prompt “green giraffe” in a “cartoon style.” In such cases, combining these generative models improves the diversity scores. We believe this scenario is common in many applications of prompt-driven generative AI.
>
> [1] Stein et al., “Exposing flaws of generative model evaluation metrics and their unfair treatment of diffusion models”, NeurIPS 2023

---

> > ### Comment · Reviewer_x3ew · 2024-11-27
> >
> > Thanks for the insightful response. I agree that the proposed algorithm better suits a "remote setting" where memory is less concerned. Overall, I'm supportive of the paper's acceptance.

---

> > > ### Author Response · Authors · 2024-11-29
> > >
> > > We thank Reviewer x3ew for the feedback on our response. We are pleased to hear that the response addressed the reviewer's comments.

---

### Author Response · Authors · 2024-11-22
**Authors' General Response**

We thank the reviewers for their constructive and thoughtful feedback. An updated paper has been submitted (changes are highlighted in blue). Here we address a common question from the reviewers. Responses to the other comments of the reviewers are posted under each review.

**1-Motivations behind the online selection of generative models**


We argue that the combination of generative models is inherently an online learning problem. Generating samples from a model is costly, in terms of computational time and resources, and perhaps monetary cost for commercial models, e.g. Dall-E and Flux 1.1 Pro. Therefore, we should naturally generate a small number of samples from each model and evaluate them, before we decide which model to use next, and keep using the new samples to guide our selection of the next model to use. This is similar to how a human would act when given a number of generative models that are costly to use. This paper proposes an algorithm to automate this process, with a tight bound on its regret (see updated discussion after Theorem 2).

An offline two-stage method (where we first generate a fixed large number of samples from each model in Stage 1, and then use them to compute the mixture distribution for the remaining samples in Stage 2) can be suboptimal since we are not discarding the obviously suboptimal arms in the middle of Stage 1, and we are not utilizing the information in the new samples in Stage 2 to update the mixture distribution.

---

### Meta-Review · Area_Chair_prmw · 2024-12-20

**Metareview:**

This paper addresses the problem of online selection to obtain a mixture of generative models.
To minimize the number of sample queries needed to form the optimal mixture model, the authors propose the Mixture Upper Confidence Bound (Mixture-UCB) online learning approach, by maximizing kernel-based evaluation scores such as the Kernel Inception Distance (KID) and Renyi Kernel Entropy (RKE).
Both theoretical results (e.g., regret bounds) and empirical results are provided to demonstrate the effectiveness of the proposed method.

Following the rebuttal phase, the reviewers reached a consensus that the (novel) problem under study is well-motivated and that the contribution of this work is significant.
As such, I recommend accepting the paper.
However, **please update** the final submission to reflect the discussions with all reviewers (and particularly Reviewers x3ew and mqps), including a discussion of the potential limitations (of the setting, and/or of the proposed approach).

**Additional Comments On Reviewer Discussion:**

The reviewers raised the following points:

- Motivation (of online selection, or to minimize the number of samples, raised by all reviewers): The authors successfully convinced the reviewers that the problem setting is both significant and of interest.
- Limitations of the current setting and/or proposed approach (raised by Reviewers x3ew and mqps): This concern was **only partially resolved** by the authors during the rebuttal phase.

I have carefully considered all of the above points in making my final decision.

---

### Decision · Program_Chairs · 2025-01-22

Accept (Poster)